# Structural and molecular basis for Cardiovirus 2A protein as a viral gene expression switch

Chris H. Hill [1,2,5,6✉], Lukas Pekarek[3,6], Sawsan Napthine[1,6], Anuja Kibe [3], Andrew E. Firth [1], Stephen C. Graham [1✉], Neva Caliskan [3,4✉] & Ian Brierley[1✉]

Programmed –1 ribosomal frameshifting (PRF) in cardioviruses is activated by the 2A protein, a multi-functional virulence factor that also inhibits cap-dependent translational initiation. Here we present the X-ray crystal structure of 2A and show that it selectively binds to a pseudoknot-like conformation of the PRF stimulatory RNA element in the viral genome. Using optical tweezers, we demonstrate that 2A stabilises this RNA element, likely explaining the increase in PRF efficiency in the presence of 2A. Next, we demonstrate a strong interaction between 2A and the small ribosomal subunit and present a cryo-EM structure of 2A bound to initiated 70S ribosomes. Multiple copies of 2A bind to the 16S rRNA where they may compete for binding with initiation and elongation factors. Together, these results define the structural basis for RNA recognition by 2A, show how 2A-mediated stabilisation of an RNA pseudoknot promotes PRF, and reveal how 2A accumulation may shut down translation during virus infection.

[1] Division of Virology, Department of Pathology, University of Cambridge, Tennis Court Road, Cambridge CB2 1QP, UK. [2] MRC Laboratory of Molecular Biology, Cambridge Biomedical Campus, Francis Crick Ave, Cambridge CB2 0QH, UK. [3] Helmholtz Institute for RNA-based Infection Research (HIRI), Helmholtz Centre for Infection Research (HZI), Würzburg, Germany. [4] Medical Faculty, Julius-Maximilians University Würzburg, Josef-Schneider-Straße 2/ D15, 97080 Würzburg, Germany. [5] Present address: Department of Biology, University of York, Wentworth Way, York YO10 5DD, UK. [6] These authors contributed equally: Chris H. Hill, Lukas Pekarek, Sawsan Napthine. ✉email: chris.hill@york.ac.uk; scg34@cam.ac.uk; neva.caliskan@helmholtz-hiri.de; ib103@cam.ac.uk

PRF is a translational control strategy employed by many RNA viruses, where it ensures the production of proteins in optimal ratios for efficient virus assembly and enables viruses to expand their coding capacity through the utilisation of overlapping ORFs (reviewed in refs. [1–3]). In canonical PRF, elongating ribosomes pause over a heptanucleotide "slippery sequence" of the form X_XXY_YYZ when they encounter a "stimulatory element" about 5–9 nucleotides downstream in the mRNA. During this time, a –1 frameshift may occur if codon-anticodon re-pairing takes place over the X_XXY_YYZ sequence: the homopolymeric stretches allow the tRNA in the P-site to slip from XXY to XXX, and the tRNA in the A-site to slip from YYZ to YYY[4–7]. A diverse array of stem-loops and pseudoknots are known to induce frameshifting, and the stability, plasticity and unfolding kinetics of these RNA elements are thought to be the primary determinants of PRF efficiency[8–10], along with the thermodynamic stability of the codon-anticodon interactions[6]. Cardioviruses present a highly unusual variation to conventional viral PRF in which the virally encoded 2A protein is required as an essential *trans*-activator[11,12]. Here, the spacing between the slippery sequence and stem-loop is 13 nt, significantly longer than typically seen, and 2A protein has been proposed to bridge this gap through interaction with the stem-loop[12]. This allows for temporal control of gene expression as the efficiency of –1 frameshifting is linked to 2A concentration, which increases with time throughout the infection cycle[12].

2A is a small, basic protein (~17 kDa; 143 amino acids; pI ~9.1) generated by 3C-mediated proteolytic cleavage at the N-terminus[13] and Stop-Go peptide release at the C-terminus[14]. Despite the identical name, it has no homology to any other picornavirus "2A" protein[15], nor any other protein of known structure. The PRF-stimulatory activity of 2A is related to its ability to bind to the RNA stimulatory element[12]. However, 2A also binds to 40S ribosomal subunits[16], inhibits apoptosis[17] and contributes to host cell shut-off by inhibiting cap-dependent translation. A C-terminal YxxxxLΦ motif has been proposed to bind to and sequester eIF4E in a manner analogous to eIF4E binding protein 1 (4E-BP1)[16], thereby interfering with eIF4F assembly[18]. However, the absence of structural data has precluded a definitive molecular characterisation of this multifunctional protein, and the mechanism by which it recognises RNA elements and stimulates frameshifting remains obscure.

Here we present the crystal structure of 2A from encephalomyocarditis virus (EMCV), revealing a novel RNA-binding fold that we term a "beta-shell". We show that 2A binds directly to the frameshift-stimulatory element in the viral RNA with nanomolar affinity and equimolar stoichiometry, and we define the minimal RNA element required for binding. Through site-directed mutagenesis and the use of single-molecule optical tweezers, we study the dynamics of this RNA element, both alone and in the presence of 2A. By observing short-lived intermediate states in real-time, we demonstrate that the EMCV stimulatory element exists in at least two conformations and 2A-binding stabilises one of these, an RNA pseudoknot, increasing the force required to unwind it. Finally, we report a direct interaction of 2A with both mammalian and bacterial ribosomes. High-resolution cryo-electron microscopy (cryo-EM) characterisation of 2A in complex with initiated 70S ribosomes reveals a multivalent binding mechanism and defines the molecular basis for RNA recognition by the 2A protein. It also reveals a likely mechanism of 2A-associated translational modulation, by competing for ribosome binding with initiation factors and elongation factors. Together, our work provides a new structural framework for understanding protein-mediated frameshifting and 2A-mediated regulation of gene expression.

## Results

**Structure of EMCV 2A reveals an RNA-binding fold.** Following recombinant expression in *E. coli*, purified 2A was analysed by SEC-MALS, revealing a predominantly monodisperse, monomeric sample (Fig. 1a, b; observed mass 18032.8 Da vs calculated mass 17930.34 Da), with a small proportion of dimers (observed mass 40836.0 Da). We crystallised the protein and determined the structure by multiple-wavelength anomalous dispersion (MAD) analysis of a selenomethionyl derivative. The asymmetric unit (ASU) of the $P6_222$ cell contains four copies of 2A related by non-crystallographic symmetry (NCS), and the structure was refined to 2.6 Å resolution (Supplementary Table 1). Unexpectedly, the four molecules are arranged as a pair of covalent 'dimers' with an intermolecular disulfide bond forming between surface-exposed cysteine residues (C111). This arrangement is likely an artefact of crystallisation, which took >30 days, possibly due to the gradual oxidation of C111 promoting formation of the crystalline lattice. The N-terminal 10–12 residues are disordered in all chains except B, in which they make long-range contacts with another chain. Similarly, C-terminal residues beyond 137 are absent or poorly ordered in all chains.

2A adopts a compact, globular fold of the form $\beta_3\alpha\beta_3\alpha\beta$ (Fig. 1c). Searches of PDBeFOLD[19], DALI[20] and CATHEDRAL[21] databases failed to reveal structural homology to any other protein, so we term this fold a "beta shell". The most striking feature is a curved, seven-stranded anti-parallel beta sheet (Fig. 1d). The concave face of the beta sheet is supported by tight packing against the two alpha helices: together, this comprises the hydrophobic core of the fold. In contrast, the solvent-exposed convex face and surrounding loops are enriched with arginine, lysine and histidine residues, conferring a strong positive electrostatic surface potential at physiological pH. Superposition of the four NCS-related chains and an analysis of the atomic displacement factors reveals regions of flexibility (Fig. 1e, f). In addition to the N- and C- termini, the β2-loop-β3 region (residues 28–37) exists in multiple conformations that deviate by up to 5.8 Å in the position of the $C_\alpha$ backbone. Similarly, the arginine-rich loop between β5 and β6 ("arginine loop", residues 93–100) is mobile, with backbone deviations of up to 4.5 Å.

Several previous studies have described mutations, truncations or deletions in EMCV 2A that affect its activity[22–24]. Many of the truncations would severely disrupt the fold and the results obtained with these mutants should be interpreted with caution. However, the loop truncation ($2A_{\Delta94-100}$) and point mutations made by Groppo et al.[23] would have only minor effects (Supplementary Fig. 1). Notably, in 2A, a C-terminal YxxxxLΦ motif predicted to bind eIF4E is within a beta strand, whereas the equivalent motif in 4E-BP1 is alpha-helical[25]. As a result, Y129 is partially buried and distal to both L134 and I135. Overlay of our 2A structure with the structure of the eIF4E:4E-BP1 complex indicates that without a significant conformational change, this motif is unlikely to represent the mechanism by which 2A recognises eIF4E (Supplementary Fig. 1).

**2A binds to a minimal 47 nt pseudoknot in the viral RNA.** The RNA sequence that directs PRF in EMCV consists of a G_GUU_UUU slippery sequence and a stimulatory stem-loop element downstream (Fig. 2a). We have previously demonstrated that three conserved cytosines in the loop are essential for 2A binding[12] (Fig. 2a). To map the interaction between 2A and the stimulatory element in more detail, we prepared a series of synthetic RNAs with truncations in the shift site, loop, and 5′ and 3′ extensions on either side of the stem (EMCV 1–6; Fig. 2b). These were fluorescently labelled at the 5′ end, and their binding to 2A

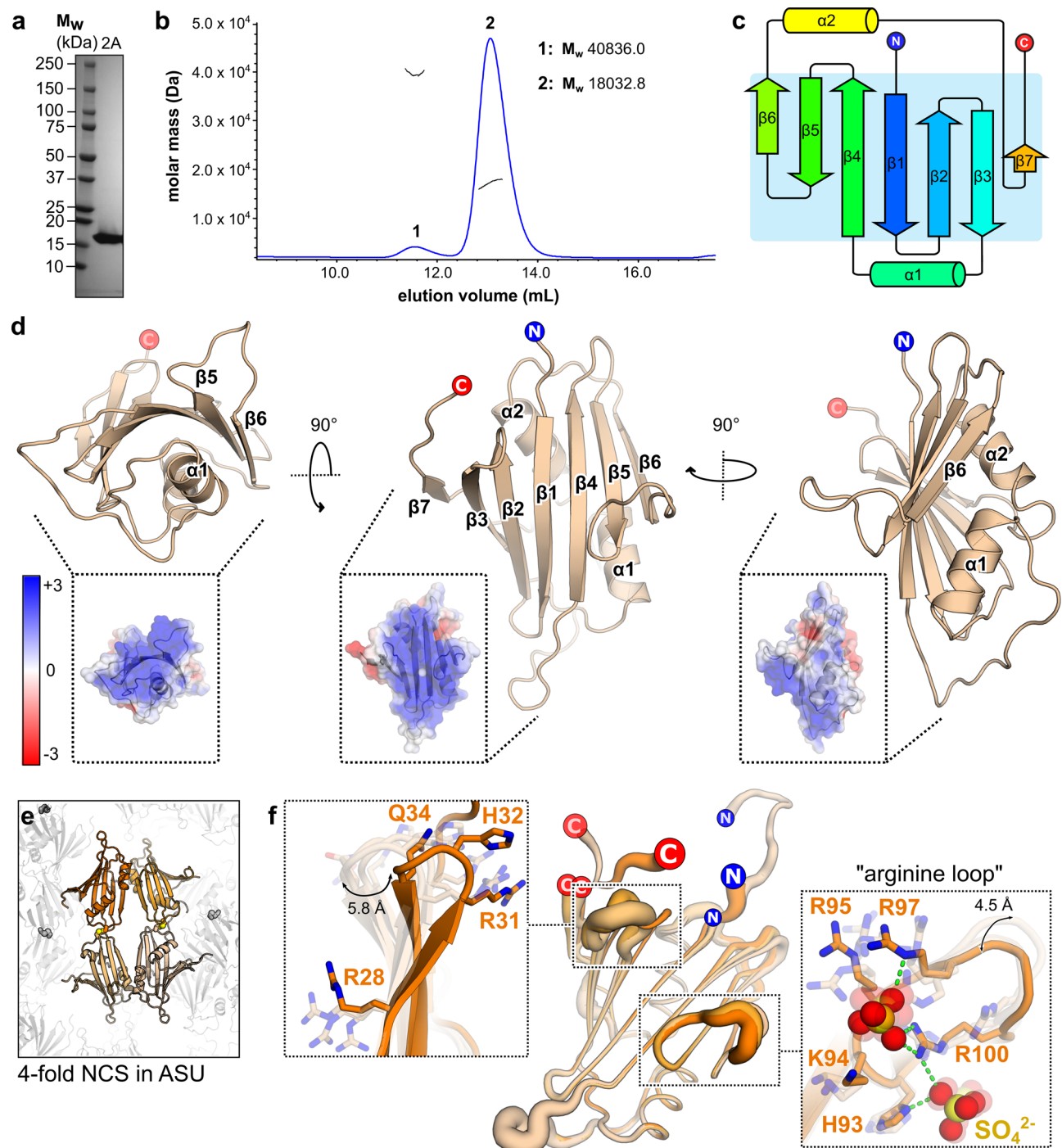

**Fig. 1 2A adopts a highly basic RNA-binding fold with intrinsic flexibility. a** SDS-PAGE analysis of EMCV 2A (Coomassie). Representative gel from five independent purifications. **b** SEC-MALS analysis of 2A. The differential refractive index is shown across the elution profile (blue) and weight-averaged molar masses of the indicated peaks are listed. **c** Topological diagram of "beta-shell" fold: a curved central sheet comprising seven antiparallel beta strands, supported by two helices. **d** Crystal structure of EMCV 2A in three orthogonal views. N- and C- termini are indicated. <Inset> Electrostatic surface potential calculated at pH 7.4, coloured between +3 (blue) and −3 (red) kT/e⁻. **e** Four molecules of 2A are present in the asymmetric unit of the crystal, arranged as two pairs of disulfide-linked dimers (spheres). **f** Superposition of the four NCS-related 2A chains in **e** reveals regions of conformational flexibility. The width of the cartoon is proportional to atomic B-factor. <Insets> Close-up view of surface loops exhibiting the greatest variation per molecule. Flexible sidechains are shown as sticks, and the Cα backbone deviation is indicated in Å. The positions of two sulfate ions from the crystallisation buffer are indicated with spheres. Source data are provided as a Source Data file.

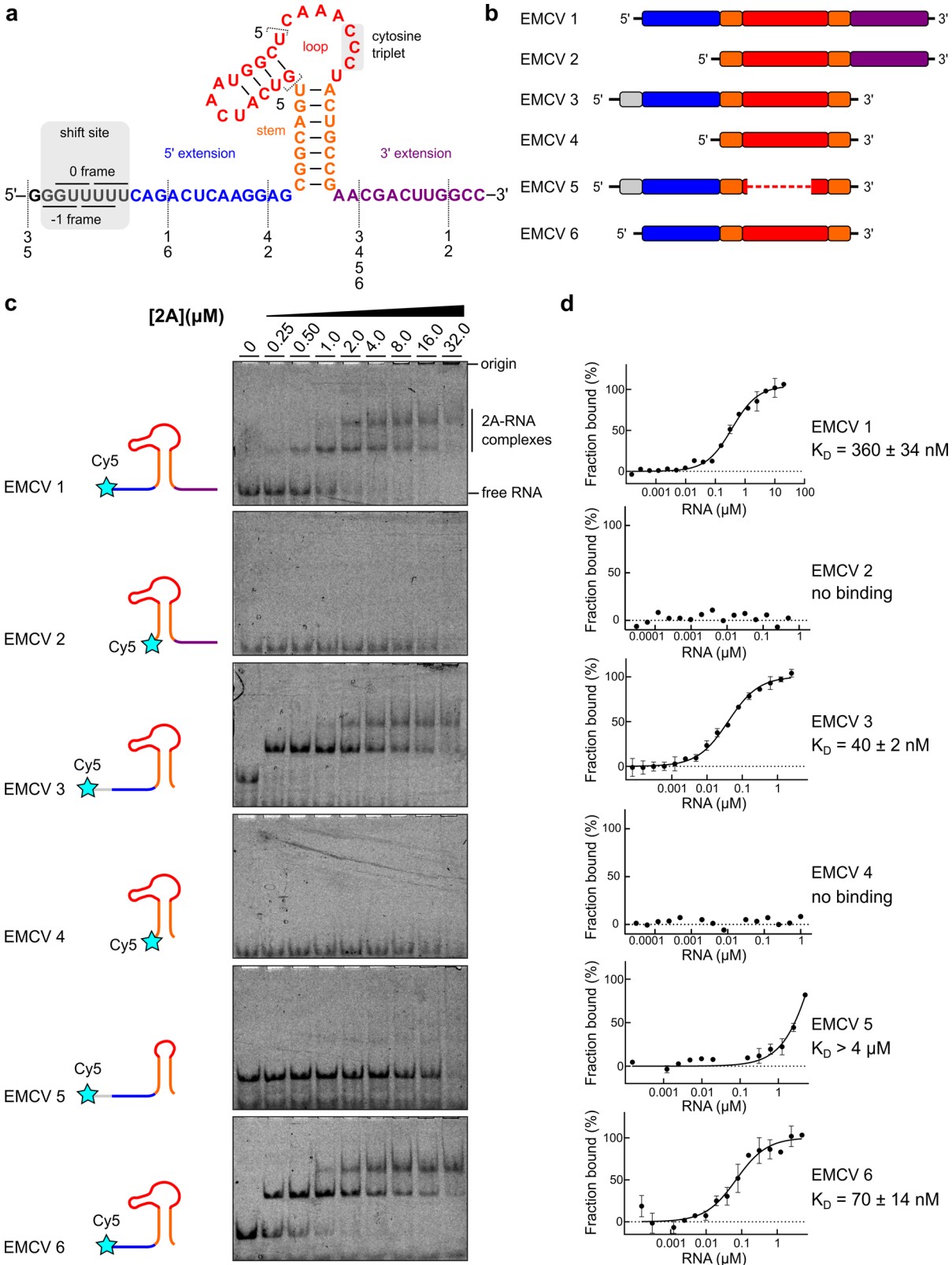

**Fig. 2 2A binds to a minimal 47 nt element in the viral RNA. a-b** Sequences and schematic diagrams of the EMCV 1–6 constructs used to assay 2A binding. **c** EMSA analyses showing that removal of the 5′ extension (blue) disables 2A binding. **d** Microscale thermophoresis (MST) was used to quantify the interactions observed in **c**. All measurements were repeated as two independent experiments and error bars represent the standard deviation from the mean. RNA concentration ranges between 60 pM–20 μM (for EMCV 1) and 150 pM–5 μM (for EMCV 2–6). Source data are provided as a Source Data file.

were analysed by electrophoretic mobility shift assay (EMSA; Fig. 2c) and microscale thermophoresis (MST; Fig. 2d, Supplementary Table 2).

Binding of 2A to EMCV 1 RNA is high affinity ($K_D = 360 \pm 34$ nM). Removal of the 3′ extension, as in EMCV 3 and EMCV 6, further increases the affinity ($K_D$ values of $40 \pm 2$ and $70 \pm 14$ nM, respectively), perhaps by removing competing base-pairing interactions. There is no substantial difference between affinities of EMCV 3 and 6, which differ only by the presence of the shift site. Removal of the 5′ extension, as in EMCV 2 and EMCV 4, completely abolishes 2A binding, and truncation of the loop, including a putative second stem (EMCV 5) reduces binding to micromolar levels. Truncating the disordered N- and C-termini of 2A, or mutating the disulfide-forming C111 residue has no effect on RNA binding (Supplementary Fig. 2). To investigate stoichiometry, we performed an isothermal titration calorimetry (ITC) analysis of the interaction between 2A and EMCV 6 (Supplementary Fig. 2). Equimolar binding was observed, with a measured $K_D$ ($246 \pm 72$ nM), similar to those obtained using MST. The dominance of enthalpy ($\Delta H$, $-13.9 \pm 0.81$ kcal/mol) to the overall free energy of binding ($\Delta G$, $-9.02$ kcal/mol) indicates an interaction mechanism driven by hydrogen bond or electrostatic contact formation. Finally, reciprocal MST experiments with fluorescently labelled 2A and unlabelled RNA yielded similar $K_D$ values (Supplementary Fig. 2, Supplementary Table 2).

We next asked whether these small RNAs could act as competitors to sequester 2A and reduce the efficiency of PRF in rabbit reticulocyte lysate (RRL) in vitro translation reactions programmed with a frameshift reporter mRNA (Supplementary Fig. 2). Indeed, when unlabelled EMCV 1, 3 and 6 were added in excess, they were able to compete with the stimulatory element present in the reporter, thereby reducing the amount of the −1 frame product. In contrast, EMCV 2, 4 and 5 had no such effect, reinforcing the results of direct binding experiments.

The failure of 2A to bind to EMCV 2, 4 and 5 was unexpected as these RNAs retain the main stem and the conserved cytosine triplet in the putative loop region. A possible explanation is that the frameshift-relevant state may include an interaction between the loop and the 5′ extension, forming a different conformation that 2A selectively recognises. To test this, we carried out mutagenesis of the 5′ extension and loop C-triplet. Individually, G7C and C37G mutations both reduce 2A-dependent PRF to near-background levels (Supplementary Fig. 3). However, in combination, the G7C + C37G double mutation restores PRF to wild-type levels, and EMSA experiments with these mutants confirm that this is due to inhibition and restoration of 2A binding. Together, this demonstrates the likelihood of a base pair between positions 7 and 37 that is necessary to form a conformation that 2A selectively recognises. Using this base pair as a restraint, RNA structure prediction[26,27] reveals a pseudoknot-like fold (Supplementary Fig. 3).

**Single-molecule measurements of stimulatory element unwinding reveal several conformations.** Information is limited in ensemble measurements of RNA-protein interactions due to molecular averaging. To further explore the effects of 2A on the unfolding and refolding of individual EMCV RNA molecules, we used optical tweezers (Fig. 3a). In force-ramp experiments, a single RNA molecule is gradually stretched and relaxed in several cycles at a constant pulling rate. The applied force allows the RNA molecule to transition between folded and unfolded states, and sudden changes in recorded force-distance (FD) trajectories indicate transitions between RNA conformations (Fig. 3c, d)[28–30]. By

mathematically fitting each FD curve (Methods) we can obtain information on the physical properties of the RNA such as the change in the contour length (maximum possible extension), which indicates whether our data are physically consistent with predicted structures of the EMCV RNA. In addition to the pseudoknot (40 nt, discussed above), *mfold*[31] suggested two other possible conformations for the frameshift stimulatory element: a stem loop (35 nt) and an extended stem-loop with additional interactions between 5′ and 3′ flanking regions (49 nt) (Fig. 3b, Supplementary Table 6). Alongside the wild-type EMCV RNA sequence (WT), we also used a mutant with a substitution in the cytosine triplet (CUC) which is known be crucial for 2A binding and PRF[12] (Fig. 3a; lower).

We initially monitored the unfolding and refolding of WT and CUC RNAs in the absence of 2A. In WT RNA, the majority of FD trajectories were characterized by a single rip at $9.3 \pm 2.3$ pN force (Fig. 3c, e). Upon release of the force, the molecules readily refolded at $6.5 \pm 3.0$ pN, showing that the process is reversible (Fig. 3c, f, Supplementary Table 3). The change in contour length calculated from the fits was $\sim 26.3 \pm 5.4$ nm (Supplementary Fig. 4, Supplementary Table 3) corresponding to a length of 46 single-stranded nucleotides. This is in close agreement with the predicted 49 nt long extended stem-loop formed by the EMCV PRF RNA (Fig. 3b, Supplementary Table 6). Interestingly, we observed similar (un)folding trajectories with the CUC RNA, with a rip occurring at $8.6 \pm 4.2$ pN and a contour length change of about $27.2 \pm 4.3$ nm (Fig. 3e, f, Supplementary Fig. 4, Supplementary Table 3), suggesting both RNAs would essentially fold into a stem-loop of similar length.

In a small fraction of WT FD trajectories (~12%) we observed a single unfolding event at higher forces above 20 pN, while refolding was unchanged ($6.5 \pm 3.0$ pN), suggesting the existence of a WT conformer with resistance to unfolding. Indeed, the putative EMCV pseudoknot would comprise 40 nucleotides, and lead to an expected difference of 23 nm in contour length upon unfolding (Supplementary Fig. 4, Supplementary Table 6). Since both the predicted pseudoknot and extended-stem loop are of similar length, and the distributions of contour length change are quite broad, this parameter is not precise enough to unambiguously distinguish between these conformations. On the other hand, the resistance to unfolding and hysteresis during refolding are well-known characteristics of more complex structures such as pseudoknots[9,32], and this is also consistent with our mutational analysis (Supplementary Fig. 3; discussed above). Furthermore, these higher force unfolding events are not observed in the CUC mutant, which is very unlikely to form a pseudoknot (Supplementary Fig. 4, Supplementary Table 3).

Next, we compared the energetics of folding and unfolding of the conformers. In optical tweezer experiments, the work of unfolding is the work required to extend the folded RNA construct (dsDNA:RNA handles and dsRNA) minus the work required to extend the fully unfolded (dsDNA:RNA handles and ssRNA) construct ($W = W_{ds} - W_{ds+ss}$). Accordingly, free energy values of WT and CUC constructs were calculated as $-13.6 \pm 4.6$ and $-14.5 \pm 4.7$ kcal/mol, respectively, which are close to the *mfold*-predicted Gibbs free energy values for the stem loop ($-14 \pm 0.7$ kcal/mol) and extended stem-loop ($-16.2 \pm 0.8$ kcal/mol) (Supplementary Table 3). This further supports the view that the EMCV WT and CUC mutant RNAs predominantly fold into the predicted stem loops.

**2A favours the formation of an alternative conformation with resistance to mechanical unwinding.** We next tested how 2A binding influences RNA stability and resistance of RNA to mechanical unwinding. For the wild-type RNA, global analysis of

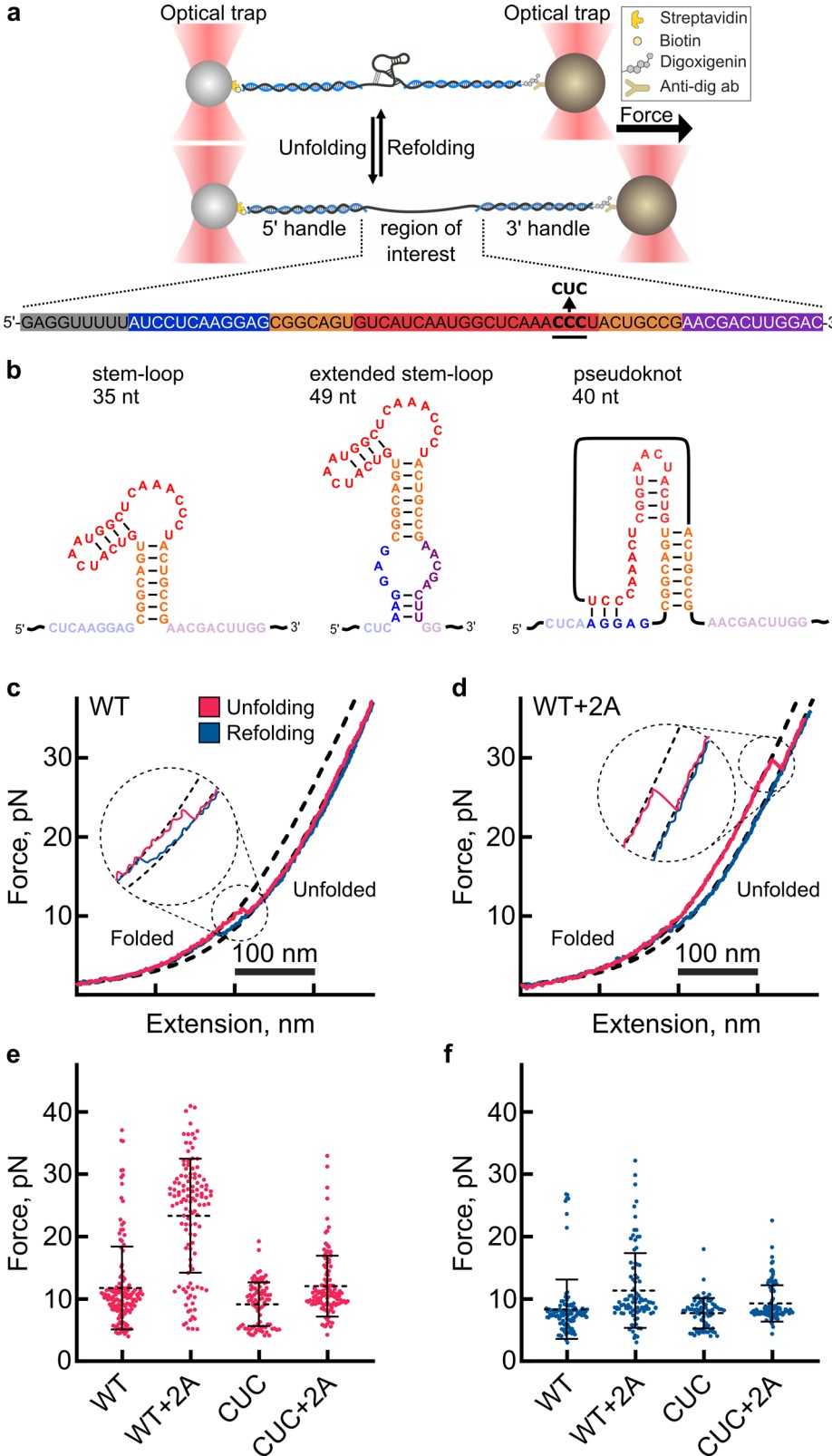

the unfolding forces reveals a 2A-induced stabilisation, which increased the fraction of unfolding events at higher forces (27.0 ± 4.2 pN) (Fig. 3d, e, Supplementary Fig. 4, Supplementary Table 3). However, refolding of the RNA was mostly unaffected (Fig. 3d, f, Supplementary Table 3). Probability distributions of the change in contour length show a peak at around 20.0 ± 3.4 nm (Supplementary Fig. 4, Supplementary Table 3) which may reflect

the unfolding of either the stem-loop (21 nm expected) or the proposed pseudoknot conformation (23 nm expected) (Fig. 3b, Supplementary Fig. 4, Supplementary Table 6).

Subsequently, we examined 2A binding to the CUC mutant RNA (Supplementary Fig. 4). In contrast to the wild-type RNA, within this population, we did not observe any stabilisation in the presence of 2A (Fig. 3e, Supplementary Fig. 4). Thus, the

**Fig. 3 Conformations of EMCV frameshifting RNA and effect of 2A on RNA unwinding. a** <Upper> Schematic diagram illustrating the optical tweezer experiments. RNA is hybridized to ssDNA handles and immobilised on beads. These are used to exert pulling force on the RNA with a focused laser beam. <Lower> Primary sequence of the construct used in optical tweezer experiments, colour coded as in Fig. 2. The location of the cytosine triplet (wild-type, WT) and point mutation (CUC) is indicated. **b** Predicted conformations of the RNA construct in **a**. The number of nucleotides involved in each folded structure is indicated. Also see Supplementary Table 6. **c** Representative force-distance curves of the unfolding (pink) and refolding (blue) transitions of the wild-type (WT) CCC RNA element. **d** Representative force-distance curves of the unfolding (pink) and refolding (blue) transitions of the wild-type (WT) CCC RNA element in the presence of 300 nM 2A protein. **e** Global analysis of all the unfolding force trajectories. Number of individual measurements are WT = 117, WT + 2A = 104, CUC = 85, CUC + 2A = 109. Data (black line) are presented as mean values ± SD error bars. **f** Global analysis of all refolding force trajectories. Number of individual measurements are WT = 111, WT + 2A = 89, CUC = 74, CUC + 2A = 97. Data (black line) are presented as mean values ± SD error bars. Source data are provided as a Source Data file.

unfolding and refolding force distributions overlap with those observed for CUC RNA in absence of 2A (Fig. 3e, f, Supplementary Fig. 4, Supplementary Table 3). We observed a small shift in the distribution of contour length changes towards lower values, which could be either due to non-specific interactions or stochastic noise. Overall, the lack of effect of 2A on the CUC RNA agreed well with the ensemble analysis of 2A:RNA interactions.

To further dissect the effect of 2A on EMCV RNAs, we calculated the work performed on the WT and CUC RNAs during (un)folding in the presence of 2A (Supplementary Fig. 4). For CUC RNA with 2A, the unfolding and refolding work distributions were largely overlapping, so the process can be considered reversible. We obtained a free energy value of −15.5 ± 5.0 kcal/mol, which is within the range of *mfold* predicted free energy values for the CUC stem-loop (−14 ± 0.7 kcal/mol) and the extended stem-loop (−16.2 ± 0.8 kcal/mol). For WT RNA with 2 A, the stabilisation effect shifted the calculated free energy to −26.5 ± 8.7 kcal/mol, thus moving the system away from equilibrium (Supplementary Table 3)[33,34]. The 2A-induced decrease in free energy of the wild-type RNA may be a combination of stabilisation induced by protein binding, and a change in RNA conformation. Taken together, our results support that 2A binding stabilises the EMCV stimulatory RNA element and increases its resistance to mechanical unwinding.

**2A interacts with the small ribosomal subunit in both eukaryotes and prokaryotes.** In addition to its role as a component of the stimulatory element, 2A has been reported to bind to 40S subunits in EMCV-infected cells[16]. To determine if the interaction of 2A with the 40S subunit can be reproduced ex vivo, we purified ribosomal subunits from native RRL and analysed 2A-subunit interactions by MST (Fig. 4a, b). Consistent with previous data, 2A forms a tight complex with 40S (apparent $K_D = 10 \pm 2$ nM) but not 60S. This apparent selectivity for the small subunit was also observed with purified prokaryotic ribosome subunits. 2A binds with very high affinity to 30S (apparent $K_D = 4 \pm 1$ nM; Fig. 4c), but not 50S (Fig. 4d). We next examined the binding of 2A to intact 70S ribosomes and to reconstituted, mRNA-bound 70S ribosomes at the initiation stage (70S IC; initiator tRNA$^{Met}$ in the P-site and an empty A-site). We were able to detect high-affinity interactions with both uninitiated and initiated 70S ribosomes (Fig. 4e, f).

**Prokaryotic ribosomes are responsive to 2A-mediated frameshifting.** Prokaryotic translation systems are well-established models for studying eukaryotic PRF signals[35,36] but it is unknown whether they can support protein-dependent PRF. To address this, we measured the efficiency of the EMCV signal in a reconstituted prokaryotic translation system and in *E. coli* S30 extracts using frameshift reporter mRNAs (Supplementary

Fig. 5). In each case, 2A-dependent PRF was observed, with ~7% of ribosomes changing frame. Mutagenesis of either the shift site or the CCC triplet disabled PRF. Shortening the length of the spacer to one more optimal for prokaryotic ribosomes (from 13 to 12 nt) doubled PRF efficiency to ~15%, comparable to that measured in eukaryotic in vitro translation systems (20%)[12]. High concentrations of 2A also had an inhibitory effect on translation, similar to that seen in eukaryotic systems.

**Cryo-EM structure of a 2A-ribosome complex reveals the structural basis for RNA recognition and translational pathology.** Having validated the use of prokaryotic ribosomes as a model system to study protein-dependent PRF, we prepared complexes between 2A and the initiated 70S ribosomes and imaged them by cryo-EM (Fig. 5a, Supplementary Table 4). After processing (Supplementary Fig. 6), the final 3D reconstruction produced a density map of 2.7 Å resolution and revealed three copies of 2A bound directly to 16S rRNA of the 30S subunit in a tripartite cluster (Fig. 5b, c). The local resolution for 2A was sufficient to allow sidechain modelling and refinement. All three 2A molecules use the same RNA-binding surface (comprising variations of R46, K48, K50, K73, K94, R95 and R97) (Fig. 5d), to recognise the ribose phosphate backbone through numerous polar and electrostatic contacts (Fig. 6a–c). We mutated this putative interaction surface (Supplementary Fig. 5) and observed reduced binding to both the stimulatory element RNA and mammalian ribosome subunits and a decreased activity in stimulating PRF in vitro. 2A$_{R95A/R97A}$ was completely functionally defective, whilst 2A$_{K73A}$ and 2A$_{R46A/K48A/K50A}$ exhibited moderate and mild effects, respectively.

By comparing the quality of both the overall density for each 2A molecule, and the side-chain density at the interaction surface, we can rank the three binding sites 2A1 > 2A2 > 2A3 in order of likely affinity. 2A1 is the most well-ordered molecule, and the 2A1 binding site on the rRNA is also the most conserved between prokaryotic and mammalian ribosomes (Supplementary Fig. 7). This is therefore likely the most physiologically relevant site, and it is possible that 2A2 and 2A3 represent lower-affinity sites (a ~40-fold molar excess of 2A was used to prepare grids). 2A1 exemplifies the critical role of the "arginine loop" (Fig. 6d). R95, R97 and R100 side chains are inserted into a ~90° junction between helices 3 and 4, forming a network of electrostatic interactions that bridge the phosphate groups on both strands. This is further stabilised by the guanidinium groups stacking against each other and exposed bases (G38) (Fig. 6d). Arginine loop residues also form polar and electrostatic contacts at the 2A2 and 2A3 interfaces (Fig. 6e, f). Whilst base-specific contacts are rare, 2A2 interacts with U485 which is normally flipped out of helix 17 (Supplementary Fig. 8). Superposition of the rRNA binding sites failed to reveal a common structural motif for RNA recognition (Supplementary Fig. 7), thus conformational

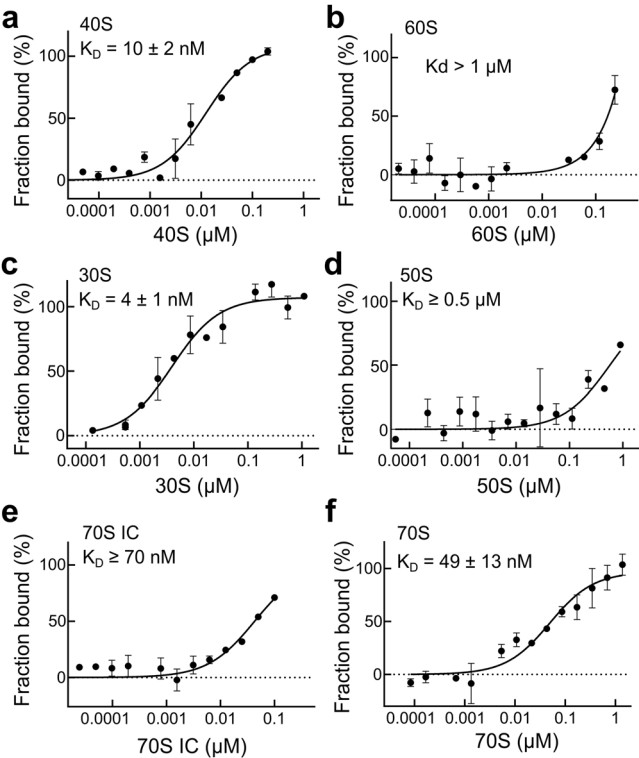

**Fig. 4 2A binds directly to eukaryotic and prokaryotic ribosomes. a** MST binding curves and apparent $K_D$ values using unlabelled 40S subunits at a concentration range of 20 pM–0.4 μM. All measurements were repeated as two independent experiments and error bars represent the standard deviation from the mean. 2A binds with high affinity to the small ribosomal subunit. **b** As in **a** with 60S subunits. Error bars as above. **c** Binding curve and apparent $K_D$ values using unlabelled 30S subunits at a concentration range of 30 pM–1 μM. Error bars as above. 2A shows a strong interaction with the prokaryotic small subunit. **d** As in **c** with 50S subunits at a concentration range of 27 pM–0.9 μM. **e** Binding curves and reported $K_D$ values for 2A-70S IC interactions. Error bars as above. **f** Same as **e**, with 2A and vacant 70S. Source data are provided as a Source Data file.

plasticity of side chains at the RNA-binding surface (Fig. 5d) explains how this protein can recognise several RNA targets. There are also intermolecular contacts between 2A protomers, consistent with our observations of multimers by SEC-MALS (Fig. 1b) and EMSA (Fig. 2c). In a subset of the data, a fourth copy of 2A (2A4) was identified to bind helix 33 of the 16S rRNA 'beak' in the 30S head. Although local resolution was only sufficient for docking, 2A4 uses the same RNA-binding surface to recognise the distorted helical backbone (Supplementary Fig. 8).

The ribosome is in an unrotated state that would normally be elongation competent, with fMet-tRNA$_i$ base-paired to the initiator codon in the P-site and mRNA available for aminoacyl tRNA delivery to the A-site[37] (Fig. 7a). There are no 2A-induced rearrangements at the decoding centre (Supplementary Fig. 8) but the presence of 2A on the 30S subunit occludes the binding site for translational GTPases. 2A1 occupies a position that would severely clash with domain II of EF-G in both compact and extended pre- and post-translocation states[38,39] (Fig. 7b). It also makes direct hydrophobic contacts with the face of S12 that would normally interact with domain III of EF-G. This 2A interaction surface on S12 is directly adjacent to the binding site for antibiotic dityromycin, which inhibits translocation by steric incompatibility with the

elongated form of EF-G[40] (Supplementary Fig. 8). 2A1 would also clash significantly with domain II of EF-Tu during delivery of aminoacyl tRNAs to the A-site[41,42] (Fig. 7c). In a similar way, 2A2 would be detrimental to both EF-G and EF-Tu binding (Fig. 7b, c). We therefore predict that 2A binding would be inhibitory to elongation and potentially initiation, via competition with IF2 during pre-initiation complex assembly[43].

## Discussion

Here we show that 2A adopts an RNA-binding fold, allowing specific recognition and stabilisation of the PRF stimulatory element in the viral RNA and direct binding to host ribosomes. Given this structural framework, we can reinterpret several preceding biochemical and virological observations. Many functions of 2A can be assigned to a single positively charged surface loop ("arginine loop" residues 93–100). Despite the low pairwise sequence identity of 2A proteins amongst Cardioviruses, R95 and R97 are completely conserved. This region was originally described as a nuclear localisation sequence (NLS)[23] and subsequently, we demonstrated that these residues are essential for PRF activity in both EMCV and Theiler's murine encephalomyelitis virus (TMEV), and that their mutation to alanine prevents 2A binding to the stimulatory element in the viral RNA[12,44] (Supplementary Fig. 5). Here we reveal how R95 and R97 also mediate direct 2A binding to the small ribosomal subunit (Fig. 6d–f) and therefore also likely confer other 2A-associated translational activities. Importantly, 2A uses the same molecular surface to bind to both the PRF stimulatory element and to ribosomes (Fig. 6d–f, Supplementary Fig. 5), so for any given 2A molecule these events are mutually exclusive. This suggests that the primary determinant of −1 PRF is likely to be 2A binding to the stimulatory element, with ribosome binding having a secondary effect. If 2A were to act as a "bridge" between the stimulatory element and the ribosome, this would necessitate two separate interaction surfaces, which we do not observe.

Our cryo-EM structure unexpectedly revealed four distinct 2A:rRNA interfaces (Fig. 6 and Supplementary Fig. 8). Based on the quality of cryo-EM density and the degree of structural conservation between prokaryotic and mammalian ribosomes, the 2A1 site is likely to be the highest affinity and most physiologically relevant (Supplementary Fig. 7). Nevertheless, all sites provide clues as to how RNA-binding specificity is achieved. RNA recognition is driven almost exclusively by electrostatic interactions between arginine or lysine side chains and the ribose phosphate backbone oxygen atoms. The mobility and flexibility of the arginine loop and other residues at the RNA binding surface (Figs. 1f and 5d) illustrate how 2A can recognise a variety of structurally degenerate targets. Whilst superposition of sites failed to reveal a common structural motif (Supplementary Fig. 7), they all include features such as kinks, distortions and junctions between multiple helices. A preference for these features is consistent with our biochemical observations that 2A is unable to bind EMCV 2, 4 and 5 RNAs, which are predicted to form stable, undistorted stem-loops (Fig. 2c, d). There is a strong likelihood that, in the 2A-bound state, the conformation of the EMCV RNA that stimulates PRF involves additional base-pairs between C-residues in the loop and a GG pair in the 5′ extension (Supplementary Fig. 3). This pseudoknot-like conformation may either pre-exist in equilibrium with other states, or it may be directly induced by 2A binding (Fig. 8). Whilst we have been unable to capture a snapshot of this molecular recognition event, it likely comprises the structural basis for the molecular "switch" that activates frameshifting during EMCV infection.

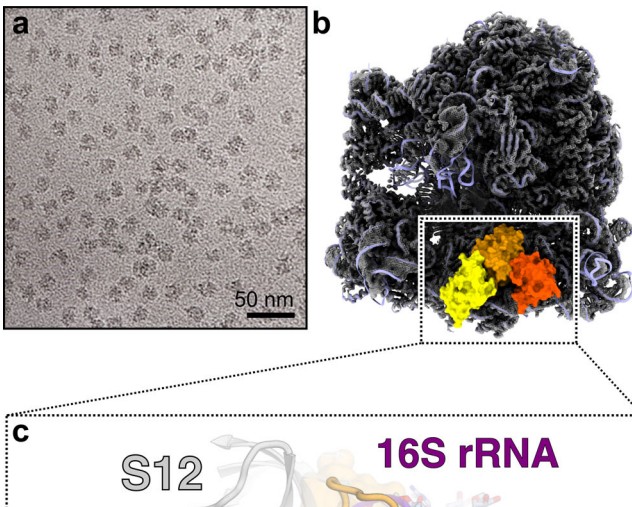

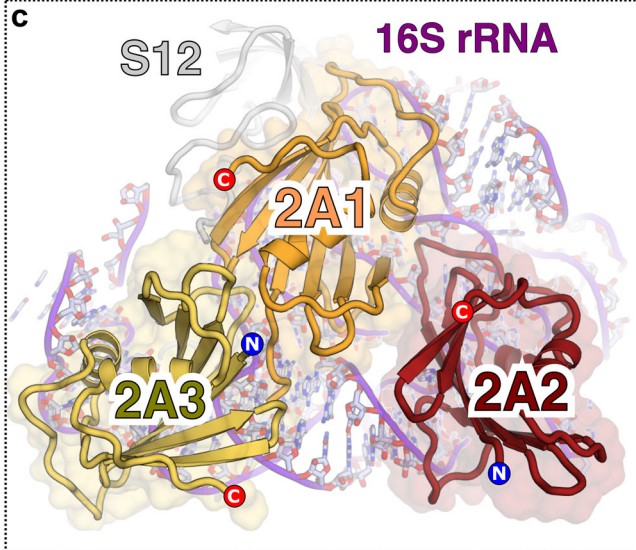

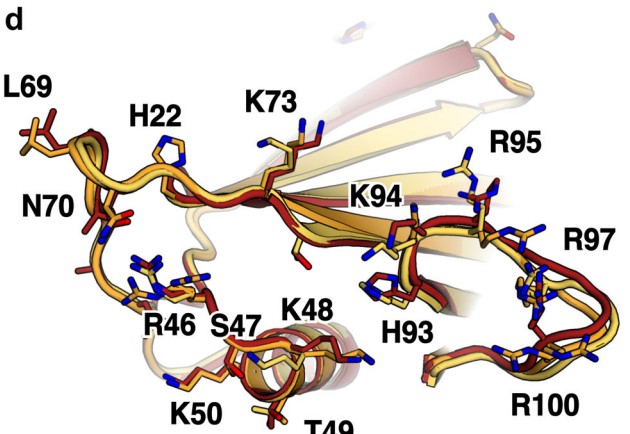

**Fig. 5 2A binds to the 70S ribosome via interactions with the 16S rRNA.**
**a** Cryo-EM analysis of a complex formed between initiated *E. coli* 70S ribosomes and EMCV 2A. Images (×75,000) were recorded on a Titan Krios microscope. Representative micrograph from dataset of 5730 images. **b** Cryo-EM map at 2.7Å resolution after focused classification and refinement. Three copies of 2A (orange, red, yellow) are bound to the 16S rRNA of the small (30S) subunit (blue ribbon). **c** Close-up view of the 2A-binding site. Ribbon diagrams of 2A (coloured as above) and ribosomal RNA (purple) are shown. Protein N- and C- termini are labelled. **d** Superposition of the three copies of 2A reveals a common RNA-binding surface with conformational flexibility. Residues involved in rRNA binding are labelled and shown as sticks.

Our single-molecule data now also provide a physical explanation for this molecular "switch". It was previously shown that ribosome can exert forces of up to 20 pN during elongation[45]. We show that, in the absence of 2A, both WT and CUC RNAs unfold at forces around ~10 pN, well within the ribosome-achievable force range and hence unlikely to cause a ribosomal pause. However, in the presence of 2A, WT but not CUC RNAs are stabilised to unwind at ~27 pN, presenting a considerable blockade to ribosome progression (Fig. 3d). This also supports the idea that the failure of the CUC mutant to stimulate PRF is due to its inability to adopt the pseudoknot-like conformation of the "switch" that would normally be selectively recognised and stabilised by 2A.

Our current mechanistic understanding of PRF is largely informed by ensemble kinetic and single-molecule FRET studies of prokaryotic ribosomes[4–6,46–48]. Frameshifting occurs late during the EF-G mediated translocation step, in which the stimulatory element traps ribosomes in a rotated or hyper-rotated state, accompanied by multiple abortive EF-G binding attempts and rounds of GTP hydrolysis. The stability of the RNA stimulatory element structure downstream of the slippery sequence is thought to be an important determinant of the frameshifting efficiency[9,49,50] although the plasticity of this structure, and the ability to adopt alternate conformations, is also a key property[10]. Several recent studies emphasise the importance of the energetics of codon:anticodon base-pairing at the slippery sequence[6,51], suggesting that the primary role of the stimulatory element is to simply pause the ribosome over a permissive slippery sequence in which the tRNA-mRNA base-pairing energies in the 0 and −1 frames are similar. Longer pauses at a more stable stimulatory element allow an equilibrium to be established between the 0 and −1 frames, converging on a maximum frameshift efficiency of ~50%. We have demonstrated how 2A-mediated stabilisation of the stimulatory element likely presents a potent elongation blockade allowing this equilibrium to be established (Figs. 3e and 8). However, this mechanism alone cannot explain the very high PRF efficiencies (up to ~70%) observed by ribosome profiling during EMCV infection[12,44].

Based on our structure, it is tempting to speculate that competition between EF-G/eEF2 and 2A1 binding might have a role in prolonging the pause, thereby contributing to the high PRF efficiencies that we observe in 2A-dependent systems[52]. Indeed, direct interactions between the ribosome and PRF stimulatory elements are not unprecedented, with a recent study describing how the HIV-1 stem-loop induces a pause by binding to the 70S A-site and preventing tRNA delivery[48]. The ribosome-bound form of 2A that we observe could therefore be a secondary 'enhancer' of PRF efficiency, acting synergistically with the main stimulatory element. It could also be relevant to the resolution of the elongation blockade: by providing an alternative 2A-binding surface that competes with the viral RNA, the ribosome may help to induce 2A dissociation from the stimulatory element during a pause at the PRF site. Alternatively, it may not be directly relevant to frameshifting per se, instead representing a way of interfering with host cell translation as 2A accumulates during infection.

In conclusion, this work defines the structural and molecular basis for the temporally regulated 'switch' behind the reprogramming of viral gene expression in EMCV infection (Fig. 8). At the heart of this is 2A: an RNA-binding protein with the remarkable ability to discriminate between stem-loop and pseudoknot conformers of the PRF stimulatory element. We also reveal how 2A interferes with host translation by specifically recognising distinct conformations within the ribosomal RNA. Together, this illustrates how the conformational plasticity of one

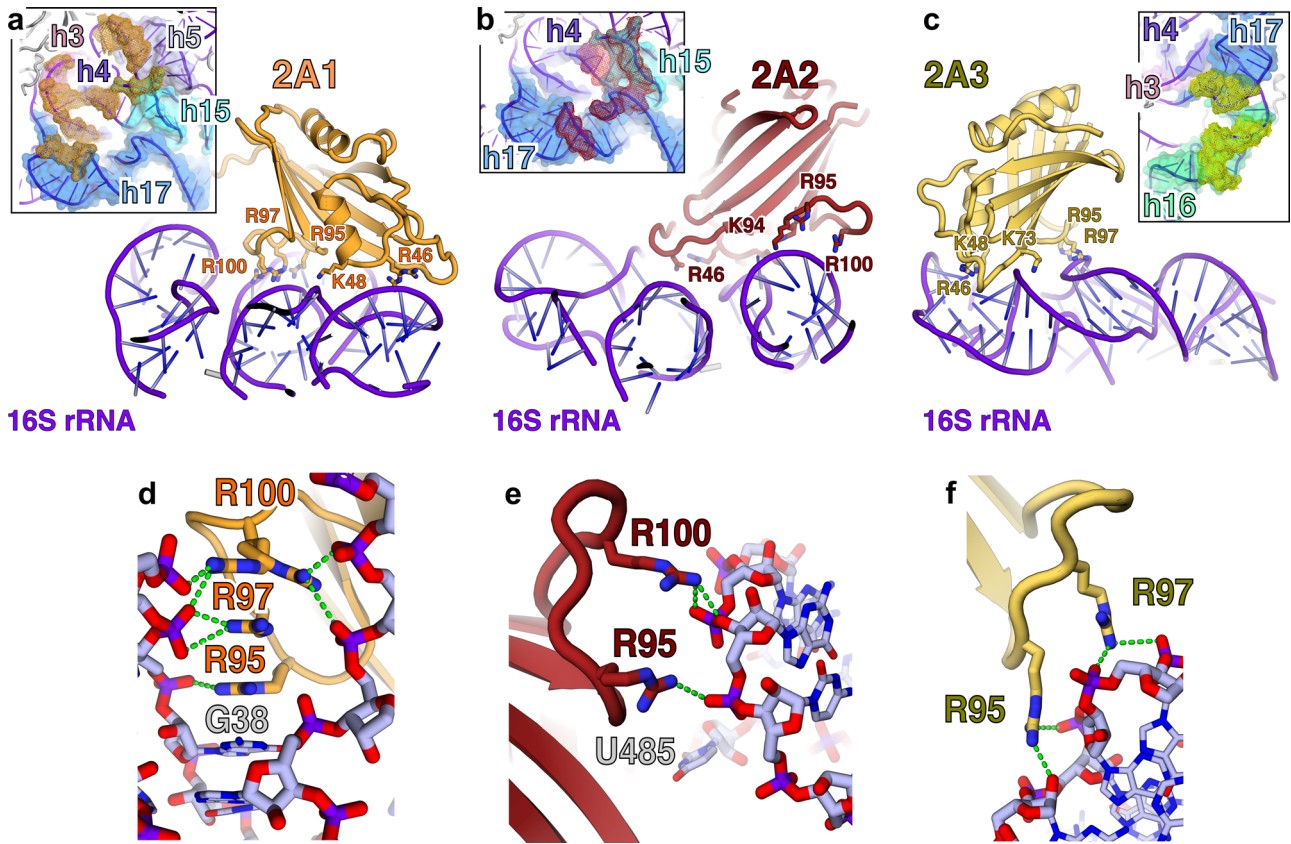

**Fig. 6 The 'arginine loop' plays a central role in RNA recognition. a–c** Details of rRNA recognition by 2A. For each copy of 2A, selected residues involved in interactions are labelled and shown as sticks. <Insets> View of the rRNA surface bound by each copy of 2A. The rRNA helices are colour-coded and labelled. The 2A contact surface is shown as a coloured mesh (orange, red and yellow, respectively). **d–f**, Close-up view of interactions between the 2A 'arginine loop' residues (R95, R97 and R100) and the rRNA backbone (sticks) for each copy of 2A (orange, red, yellow). Polar or electrostatic contacts are indicated by a green dashed line.

RNA-binding surface can contribute to multiple functions through finely tuned relative affinities for different cellular targets.

## Methods

**Materials availability**. Further information and requests for resources should be directed to and will be fulfilled by Ian Brierley (ib103@cam.ac.uk). Plasmids generated in this study are available on request. DNA and RNA oligonucleotides are standard synthetic products that are commercially available (see Supplementary Table 5).

**Cloning, protein expression and purification**. All gene cloning, manipulation and plasmid propagation steps involving pGEX6P1 or pOPT vectors were carried out in *Escherichia coli* DH5α cells grown at 37 °C in 2 × TY or LB media supplemented with appropriate selection antibiotics. EMCV 2A cDNA was amplified by PCR from previously described plasmid 2A_pGEX6P1[12] (primers E2A_F1 and E2A_R1; Supplementary Table 5) and cloned into pOPTnH[53] using NdeI and BamHI sites, thereby introducing a C-terminal GlySerLysHis₆ tag. The 2A₉₋₁₃₆ truncated construct was cloned in an identical way (primers E2A_F2 and E2A_R2; Supplementary Table 5). The EMCV 2A R95A/R97A mutant was cloned into pOPTnH after PCR-amplification from a previously described 2A_pGEX6P1 construct containing these mutations[12]. Other EMCV 2A mutants were prepared by PCR mutagenesis, using either the wild-type EMCV 2A_pOPT or 2A₉₋₁₃₆_pOPT plasmids as templates, with the following primer pairs (C111S: E2A_mut_F1 and E2A_mut_R1; R46A/K48A/K50A: E2A_mut_F2 and E2A_mut_R2; K74A: E2A_mut_F3 and E2A_mut_R3; Supplementary Table 5). To introduce an N-terminal StrepII-tag (SII-2A), annealed oligonucleotides encoding the StrepII-tag (SII_F and SII_R, Supplementary Table 5) were inserted in-frame at the BamHI site of 2A_pGEX6P1.

Recombinant proteins 2A, 2A₉₋₁₃₆; C111S, 2A_R95A/R97A, 2A_R46A/K48A/K50A and 2A_K73A were expressed in *E. coli* BL21 (DE3) pLysS cells grown in 2 × TY broth supplemented with 100 μg/mL ampicillin and 12.5 μg/mL chloramphenicol (37 °C, 200 rpm) until an OD₆₀₀ₙₘ of 0.6–1.0 was reached. Expression was induced with 0.5 mM IPTG for either 4 h at 37 °C or overnight at 21 °C. For selenomethionyl derivatisation (2A_SeMet), protein was expressed in *E. coli* B834 cells, grown shaking

(210 rpm, 37 °C) in SeMet base media (Molecular Dimensions) supplemented with nutrient mix, 40 μg/mL L-selenomethionine and 100 μg/mL ampicillin. Expression was induced as above.

Cells were harvested by centrifugation (4000 × *g*, 4 °C, 20 min), washed once in ice-cold PBS and stored at −20 °C. Pellets from four litres of culture were resuspended in cold lysis buffer (50 mM Tris-HCl pH 8.0, 500 mM NaCl, 30 mM imidazole, supplemented with 50 μg/mL DNase I and EDTA-free protease inhibitors) and lysed by passage through a cell disruptor at 24 kPSI (Constant Systems). Lysate was cleared by centrifugation (39,000 × *g*, 40 min, 4 °C) prior to incubation (1 h, 4 °C) with 4.0 mL of Ni-NTA agarose (Qiagen) pre-equilibrated in the same buffer. Beads were washed in batch four times with 200 mL buffer (as above, but without DNase or protease inhibitors) by centrifugation (600 × *g*, 10 min, 4 °C) and re-suspension. Washed beads were pooled to a gravity column prior to elution over 10 column volumes (CV) with 50 mM Tris-HCl pH 8.0, 150 mM NaCl, 300 mM imidazole. Fractions containing 2A were pooled and dialysed (3 K molecular weight cut-off (MWCO), 4 °C, 16 h) against 1 L buffer A (50 mM Tris-HCl pH 8.0, 400 mM NaCl, 5.0 mM DTT) before heparin-affinity chromatography to remove contaminating nucleic acids. Samples were loaded on a 10 mL HiTrap Heparin column (GE Healthcare) at 2.0 mL/min, washed with two CV of buffer A and eluted with a 40% → 100% gradient of buffer B (50 mM Tris-HCl pH 8.0, 1.0 M NaCl, 5.0 mM DTT) over 10 CV. Fractions containing 2A were pooled and concentrated using an Amicon® Ultra centrifugal filter unit (10 K MWCO, 4,000 × *g*). Size exclusion chromatography was performed using a Superdex 75 16/600 column pre-equilibrated in 10 mM HEPES pH 7.9, 1.0 M NaCl, 5.0 mM DTT. Purity was judged by 4-20% gradient SDS-PAGE, and protein identity verified by mass spectrometry. Purified protein was used immediately or was concentrated as above (~7.0 mg/mL, 390 μM), snap-frozen in liquid nitrogen and stored at −80 °C. Variants of 2A, including 2A₉₋₁₃₆;C111S and 2A_SeMet were purified identically to the wild-type protein. The StrepII-tagged variant (SII-2A) was expressed and purified using GST-affinity as previously described[12]. Following removal of the GST tag by 3C protease, SII-2A was further purified by Heparin affinity and size-exclusion chromatography as above.

**Size-exclusion chromatography coupled to multi-angle light scattering (SEC-MALS)**. Per experiment, 100 μL of protein was injected onto a Superdex 75

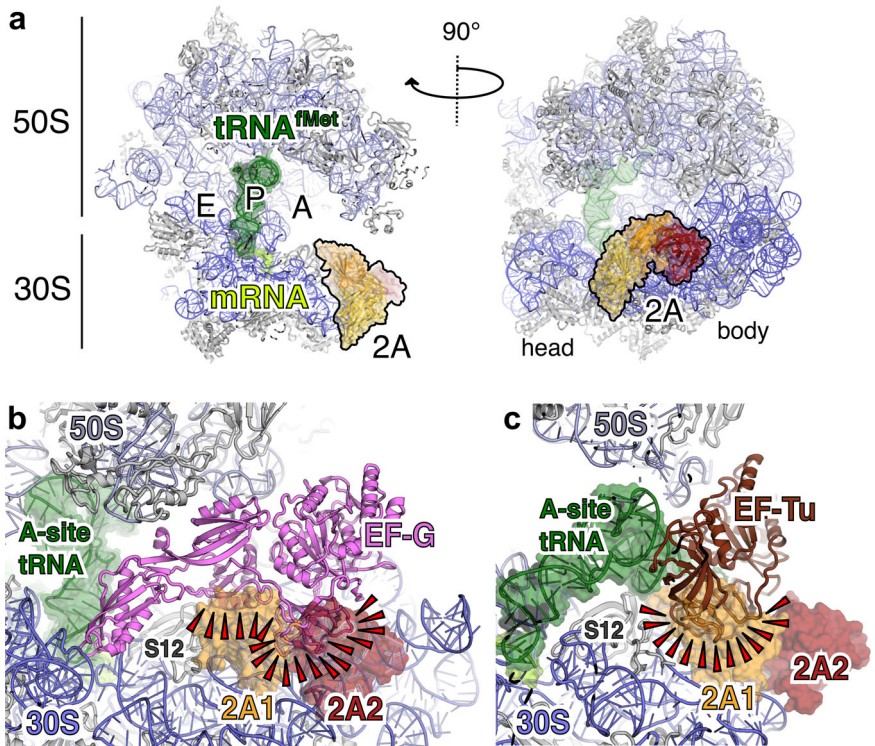

**Fig. 7 2A binding may clash with translational GTPases. a** Ribbon diagram of initiated 70S-mRNA-tRNA^fMet-2A complex. Ribosome sites are labelled A, P and E. The initiator tRNA^fMet (dark green), mRNA (light green), and 2A (orange, red, yellow) are shown in two orthogonal views. **b** Comparison of 70S-2A complex to 70S pre-translocation complex with EF-G (4V7D [10.2210/pdb4v7d/pdb]). 2A binding would clash (red wedges) with EF-G binding. **c** Comparison of 70S-2A complex to 70S complex with EF-Tu (5WE6 [10.2210/pdb5we6/pdb]). 2A binding would clash (red wedges) with EF-Tu binding.

increase 10/300 GL column (GE Healthcare) pre-equilibrated with 20 mM Tris-HCl, 1.0 M NaCl (0.4 mL/min flow, 25 °C). Experiments were performed with 5.2 mg/mL 2A (corresponding to a molar concentration of 290 μM). The static light scattering, differential refractive index, and the UV absorbance at 280 nm were measured in-line by DAWN 8+ (Wyatt Technology), Optilab T-rEX (Wyatt Technology), and Agilent 1260 UV (Agilent Technologies) detectors. The corresponding molar mass from each elution peak was calculated using ASTRA 6.1 software (Wyatt Technology).

**Protein crystallization**. Purified EMCV 2A was concentrated to 5.9 mg/ml in 10 mM HEPES pH 7.9, 1.0 M NaCl, 2.0 mM DTT. Diffraction-quality native 2A crystals were grown at 21 °C by sitting-drop vapour diffusion against an 80 μL reservoir of 0.625 M $(NH_4)_2SO_4$, 0.15 M tri-sodium citrate pH 5.7. Notably, crystal growth was only visible after 30 days. Drops were prepared by mixing 200 nL protein and 200 nL crystallization buffer. Selenomethionyl derivative 2A ($2A_{SeMet}$) was concentrated to 5.7 mg/mL in 10 mM HEPES pH 7.9, 1.0 M NaCl, 2.0 mM DTT, and diffraction-quality $2A_{SeMet}$ crystals were grown as above against an 80 μL reservoir of 0.675 M $(NH_4)_2SO_4$, 0.15 M tri-sodium citrate pH 5.7. Crystals were cryo-protected by the addition of 0.5 μL crystallization buffer supplemented with 20% v/v glycerol, prior to harvesting in nylon loops and flash-cooling by plunging into liquid nitrogen.

**X-ray data collection, structure determination, refinement and analysis**. Native datasets (Supplementary Table 1) of 900 images were recorded at Diamond Light Source, beamline I03 ($λ = 0.9796$ Å) on a Pilatus 6 M detector (Dectris), using 100% transmission, an oscillation range of 0.2° and an exposure time of 0.04 s per image. Data were collected at a temperature of 100 K. Data were processed with the XIA2[54] automated pipeline, using XDS[55] for indexing and integration, and AIMLESS[56] for scaling and merging. Crystallographic calculations were performed using the default software parameters unless otherwise stated. Processing and refinement statistics are detailed in Supplementary Table 1. Resolution cut-off was decided by a $CC_{1/2}$ value ≥ 0.5 and an $I/σ(I)$ ≥ 1.0 in the highest resolution shell[57]. For MAD phasing experiments, selenomethionyl derivative datasets were ecorded at beamline I03 (peak $λ = 0.9796$ Å, 12656.0 eV; hrem $λ = 0.9763$, 12699.4 eV; inflexion $λ = 0.9797$, 12655.0 eV). Data were processed as above using XIA2, XDS and AIMLESS. The structure was solved by three-wavelength anomalous dispersion analysis of the selenium derivative (space group $P6_222$) performed using the autoSHARP pipeline[58], implementing SHELXD[59] for substructure determination, SHARP for heavy-atom refinement and phasing, SOLOMON[60] for density modification and ARP/wARP[61] for automated model building. This was successful

in placing 503/573 (87%) residues in the ASU, which comprised four copies of the protein related by NCS. This initial model was then used to solve the native dataset by molecular replacement with Phaser[62]. The model was completed manually by iterative cycles of model building using COOT 0.9.2[63] and refinement with phenix.refine[64] (Phenix build 1.18.1_3865), using local NCS restraints and one TLS group per chain. Upon completion of model building, ISOLDE 1.1[65] was used to improve model geometry and resolve clashes prior to a final round of refinement using phenix.refine. MolProbity[66] was used throughout the process to evaluate model geometry. For the electrostatic potential calculations, partial charges were first assigned using PDB2PQR[67], implementing PROPKA to estimate protein pKa values. Electrostatic surfaces were then calculated using APBS[68]. Prior to the designation of the "beta shell" as a new fold, structure-based database searches for proteins with similar folds to EMCV 2A were performed using PDBeFOLD[19], DALI[20] and CATHEDRAL[21]. Buried surface areas were calculated using PDBePISA[69].

**RNA folding prediction**. The simRNAweb server[26] was used for stem-loop and pseudoknot tertiary structure modelling of the EMCV stimulatory element. Experimentally determined base pairs were input as secondary structure restraints. Replica exchange Monte Carlo (REMC) simulated-annealing was performed with 10 replicas and 16,000,000 iterations per cycle. Trajectory files from eight independent simulations were concatenated and clustered, and all-atom PDB files was generated from the lowest energy state in each of the five most populous clusters. The 3D models presented (Supplementary Fig. 3) represent the top cluster for pseudoknots and the top three clusters for stem-loops.

**Electrophoretic mobility shift assay (EMSA)**. Synthetic RNA oligonucleotides (Supplementary Table 5, IDT) were dissolved in distilled water. RNAs were labelled at the 5′ end with A647-maleimide or Cy5-maleimide conjugates (GE Healthcare) using the 5′ EndTag kit (Vector Labs) as directed by the manufacturer. For each binding experiment, a series of reactions were prepared on ice, each containing 1.0 μL 500 nM RNA, 1.0 μL serially diluted protein at concentrations of 320, 160, 80, 40, 20, 10, 5.0, and 2.5 μM in 10 mM HEPES pH 7.9, 1.0 M NaCl, 5.0 μL 2 × buffer (20 mM Tris-HCl pH 7.4, 80 mM NaCl, 4.0 mM magnesium acetate 2.0 mM DTT, 10% v/v glycerol, 0.02% w/v bromophenol blue, 200 μg/mL porcine liver tRNA, 800 U /mL SUPERase-In [Invitrogen]) and 3.0 μL distilled water. This gave final binding reactions of 10 μL with 50 nM RNA, 1 × buffer, a salt concentration of ~140 mM and proteins at concentrations of 32, 16, 8.0, 4.0, 2.0, 1.0, 0.5 and 0.25 μM. Samples were incubated at 37 °C for 20 min prior to analysis by native 10% acrylamide/TBE PAGE (25 min, 200 V constant). Gels were scanned with a

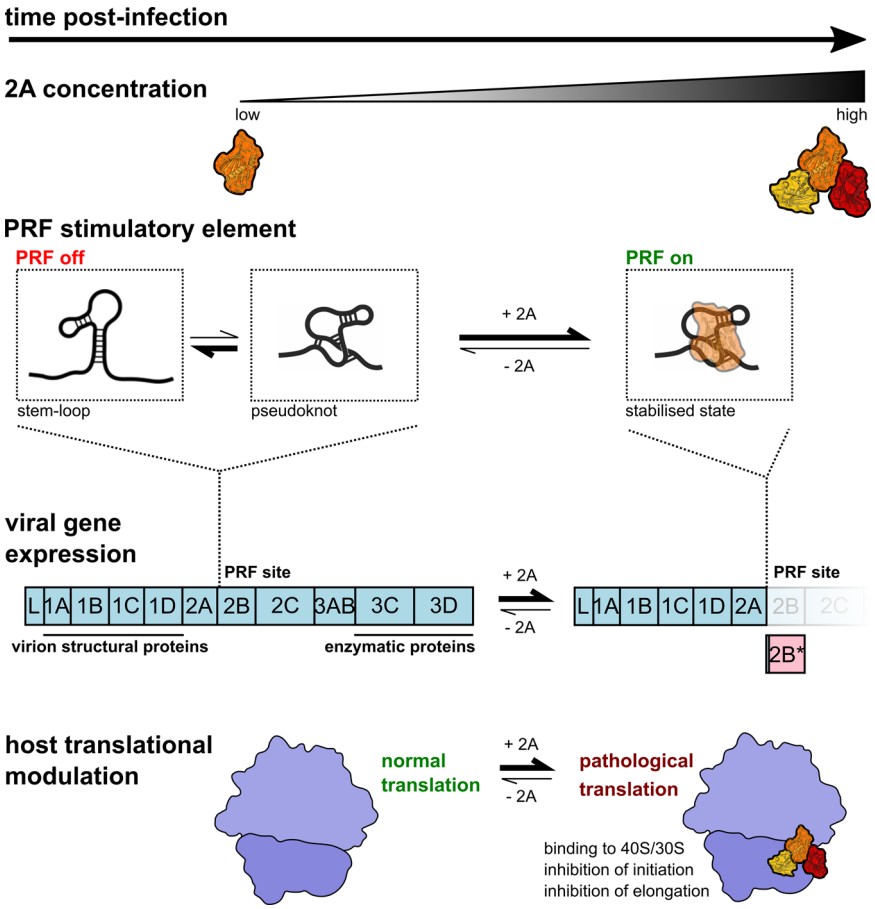

**Fig. 8 Molecular basis for 2A-induced reprogramming of gene expression.** The PRF stimulatory RNA element is predicted to form either stem-loop or pseudoknot conformations. As 2A accumulates during EMCV infection, it selectively binds to and stabilises a pseudoknot-like conformation of the PRF stimulatory element, thereby enabling PRF, producing *trans*-frame product 2B* and downregulating the expression of enzymatic viral proteins later in infection. 2A also binds directly to the small ribosomal subunit at the translational GTPase factor binding site, progressively inhibiting both initiation and elongation as it accumulates. This may contribute to the shutdown of host cell translation during lytic infection.

Typhoon FLA-7000 (GE) using the 635 nm laser / R670 filter. Raw, uncropped image data is available in the Source Data file.

**Isothermal titration calorimetry (ITC)**. ITC experiments were performed at 25 °C using an automated MicroCal PEAQ-ITC platform (Malvern Panalytical). Proteins and synthetic RNA oligonucleotides (IDT) were dialysed extensively (24 h, 4 °C) into buffer (50 mM Tris-HCl pH 7.4, 400 mM NaCl) prior to experiments. RNA (52 µM) was titrated into protein (5 µM) with $1 \times 0.4$ µL injection followed by $12 \times 3.0$ µL injections. Control titrations of RNA into buffer, buffer into protein and buffer into buffer were also performed. Data were analysed using the MicroCal PEAQ-ITC analysis software 1.30 (Malvern Panalytical) and fitted using a one-site binding model. Presented traces were representative of two independent titrations.

**Microscale thermophoresis (MST)**. For RNA-binding experiments, synthetic EMCV RNA variants (Supplementary Table 5) were dissolved in distilled water and labelled at the 5′ end with Dylight 650 maleimide conjugates (Thermo Scientific) using the 5′ EndTag kit (Vector Labs) as directed by the manufacturer. For each binding experiment, RNA was diluted to 10 nM in MST buffer (50 mM Tris-HCl pH 7.8, 150 mM NaCl, 10 mM MgCl$_2$, 2 mM DTT supplemented with 0.05% Tween 20) and a series of 16 tubes with 2A dilutions were prepared on ice in MST buffer, producing 2A ligand concentrations ranging from 0.00015 to 5 µM for EMCV RNA 2-6 and 0.00006 to 20 µM for EMCV RNA1. For the measurement, each ligand dilution was mixed with one volume of labelled RNA, which led to a final concentration of 5.0 nM labelled RNA. The reaction was mixed by pipetting, incubated for 10 min followed by centrifugation at $10,000 \times g$ for 10 min. Capillary forces were used to load the samples into Monolith NT.115 Premium Capillaries (NanoTemper Technologies). Measurements were performed using a Monolith NT.115Pico instrument (NanoTemper Technologies) at an ambient temperature of 25 °C. Instrument parameters were adjusted to 5% LED power, medium MST power and MST on-time of 10 s. An initial fluorescence scan was performed across

the capillaries to determine the sample quality and afterwards 16 subsequent thermophoresis measurements were performed. To determine binding affinities, data of at least two independently pipetted measurements were analysed for the fraction bound (MO.Affinity Analysis software, NanoTemper Technologies). For the non-binders, since the maximum amplitude would numerically be zero, deltaFnorm values were divided by the average maximum amplitude of the dataset to plot fraction bound. Data were fitted to the Kd model using MO.Affinity Analysis software (NanoTemper) and were plotted using Prism 8.0.2 (GraphPad).

Conjugation of a fluorescent label to the surface-exposed cysteine residue (C111) observed in the 2A crystal structure (Fig. 1e) provided a convenient way of studying binding to multiple unlabelled targets by MST, in such a way that the observed affinities would be directly comparable. For this experiment, EMCV 2A protein was labelled using the Protein Labelling Kit RED-Maleimide (NanoTemper Technologies) according to the manufacturer's instructions. In brief, 2A protein was diluted in a buffer containing 10 mM HEPES pH 7.9, 1.0 M NaCl and dye was mixed at a 1:3 molar ratio at room temperature for 30 min in the dark. Unreacted dye was removed on a spin gel filtration column equilibrated with 10 mM HEPES pH 7.9, 1.0 M NaCl. The labelled 2A protein was diluted to 10 nM in MST buffer. Synthetic EMCV RNA variants were used in dilutions ranging from 0.0008 to 26 µM for RNA 1 and 0.00003 to 1 µM for RNA 2-6. For the measurement, each RNA ligand dilution was mixed with one volume of labelled protein 2A, which led to a final concentration of protein 2A of 5.0 nM. Similar experiments were conducted with ribosomes in MST buffer, with ligand concentrations ranging between 0.00002 and 0.4 µM for 40S and 60S, 0.00003 and 1 µM for 30S, 0.000027 and 0.9 µM for 50S, 0.0008 and 1.375 µM for empty 70S and 0.000003 and 0.1 µM for 70S IC. The measurements were performed as described above.

**Preparation of constructs for optical tweezer experiments**. DNA encoding the frameshifting sequence of EMCV was inserted into plasmid pMZ_lambda_OT using PCR and subsequent Gibson assembly. This plasmid contains the ColE1 origin, ampicillin resistance, ribosome binding site and two 2 kbp handle regions

derived from lambda phage DNA (5′ and 3′ handle). For the generation of the mutant plasmid, PCR and blunt-end ligation was used to mutate the CCC triplet in the EMCV stem-loop to CUC. Control constructs (see below) were prepared the same way as mutant constructs. For the control construct without any single-stranded RNA region, a PCR reaction using the EMCV wild-type (CCC) construct as template was conducted with 3′ handle forward oligonucleotide and 5′ handle reverse oligonucleotide as primers (Supplementary Table 5) After the PCR, the linear products were blunt-end ligated to yield the control constructs. Wild-type and mutant plasmids were subsequently used to generate construct suitable for optical tweezer measurements consisting of the EMCV frameshifting sequence flanked by the 2 kbp long handle regions. Three pairs of primers for PCR were designed allowing the amplification of the in vitro transcription template and 5′ and 3′ handles. Subsequently, PCR reactions generated 5′ and 3′ handles and a long template for in vitro transcription. The 3′ handle was labelled during PCR using a 5′ digoxigenin-labelled reverse primer. The 5′ handle was labelled with Biotin-16-dUTP at the 3′ end following PCR using T4 DNA polymerase. RNA was transcribed from templates for in vitro transcription using T7 RNA polymerase. RNA and both DNA handles (5′ and 3′) were annealed together in a mass ratio 1:1:1 (5 µg each) by incubation at 95 °C for 10 min, 62 °C for 1 h, 52 °C for 1 h and slow cooling to 4 °C in a buffer containing 80% formamide, 400 mM NaCl, 40 mM HEPES, pH 7.5, and 1 mM EDTA[70]. Following annealing, the samples were concentrated by ethanol precipitation, the pellets resuspended in 40 µL RNase-free water, split into 4 µL aliquots and stored at –20 °C.

**Optical tweezers data collection and analysis**. Optical tweezer experiments were performed using a commercial dual-trap instrument equipped with a microfluidics system (C-trap, Lumicks). Optical tweezers (OT) constructs described above were mixed with 3 µL of polystyrene beads coated with antibodies against digoxigenin (0.1% v/v suspension, Ø 1.76 µm, Lumicks), 8 µL of measurement buffer (20 mM HEPES, pH 7.6, 300 mM KCl, 5 mM MgCl₂, 5 mM DTT and 0.05% Tween) and 1 µL of RNase inhibitors. The mixture was incubated for 20 min at room temperature in a final volume of 16 µL, and subsequently diluted by the addition of 0.5 mL measurement buffer. Separately, 0.8 µL of streptavidin (SA)-coated poly-styrene beads (1% v/v suspension, Ø 2 µm, Lumicks) was supplemented with 1 mL of measurement buffer, the flow cell was washed with the measurement buffer and suspensions of both SA beads as well as the complex of OT construct with anti-digoxigenin (AD) beads were introduced into the flow cell. Per experiment, an AD bead and a SA bead were optically trapped and brought into close proximity to allow the formation of a tether in between. The beads were moved apart (unfolding) and back together (refolding) at constant speed (0.05 µm/s) to yield the FD curves. The stiffness was maintained at 0.31 and 0.24 pN/nm for trap 1 (AD bead) and trap 2 (SA bead), respectively. For experiments with 2A protein experiments, the protein was diluted to 300 nM in measurement buffer and added to the buffer channel of the optical tweezer flow cell. FD data was recorded at a rate of 78,000 Hz. To ensure that the observed effects were indeed a result of interaction with the studied RNA region and not a non-specific binding to handle regions, we also employed constructs containing no single-stranded RNA sequence (No ssRNA control) [https://doi.org/10.17632/gkpwngy65h.2]. No oxygen scavengers were used during measurements. However, to prevent oxygen damage, all buffers were degassed and contained DTT as reducing agent.

Afterwards, the data were down sampled by a factor of 30 and filtered with a Butterworth filter (0.05 filtering frequency, filter order 4) using a custom-written python algorithm. FD curves were fitted using a custom-written Python script, which is based on Pylake package provided by Lumicks (https://lumicks-pylake.readthedocs.io/). Scripts have been deposited to GitHub [https://github.com/REMI-HIRI/EMCV_2A_project]. The fitting procedure was done as described[71]. In brief, first, a fully folded part (until the first detectable unfolding step) was fitted with a worm-like chain model (WLC)[72,73] to determine the persistence length (dsL$_P$) of the tether while the contour length (dsL$_C$) parameter was held fixed at 1256 nm (±1%; 4110 bp*0.305 nm/bp and 4 ss*0.59 nm/ss)[74]. The (partially) unfolded parts of FD curve were then fitted by a model comprising of WLC (describing the folded double-stranded handles) and a second WLC model (describing the unfolded single-stranded parts)[72,75]. For fitting of the unfolded regions, parameters extracted from the fully folded part fitting (dsL$_P$, dsL$_C$, dsK) were used and fixed in the WLC part of the combined model. The persistence length of the single-stranded part (ssL$_P$) was fixed at 1 nm while contour length (ssL$_C$) of the single-stranded part together with the single-stranded stretch modulus (ssK) were optimized. The work performed on the structure while unfolding or refolding was calculated as the difference between the area under the curve (AUC) of the fit for the folded region and AUC of the fit for the unfolded region, counted from the beginning of the FD curve till the unfolding step coordinates[76]. If the unfolding and refolding work distributions were overlapping, Crook's fluctuation theorem was applied to estimate the equilibrium work, which represents free Gibbs energy[76], as intersection between the unfolding and refolding work distributions. Since Crooks fluctuation theorem directly averages work values of unfolding and folding, it is not reliable when the system is far from equilibrium or in other cases when folding and unfolding work distributions are very different[33]. In our WT + 2A samples, the overlap between folding and unfolding work was not sufficient. Therefore, to more accurately estimate the free energies where large fluctuations exist in work distributions, we applied the Jarzynski's equality as described[34]. We then corrected for the bias in the

Jarzynski estimate[77]. Theoretical values of the Gibbs free energies for the predicted RNA structures were obtained using mfold[31]. The FD curves were plotted using Prism 8.0.2 (GraphPad). The RNAstructure software (version 6.2) was also used for prediction of the EMCV RNA element secondary structure[78].

**Eukaryotic ribosomal subunit purification**. 40S and 60S subunits were purified from untreated rabbit reticulocyte lysate (Green Hectares) as previously described[79]. Briefly, ribosomes were pelleted by centrifugation (4 °C, 270,000 × g, 4.5 h) and resuspended in 20 mM Tris-HCl pH 7.5, 4.0 mM MgCl₂, 50 mM KCl, 2.0 mM DTT. Following treatment with 1.0 mM puromycin and addition of KCl to 0.5 M, 40S and 60S subunits were separated by centrifugation (4 °C, 87,000 × g, 16 h) through a sucrose density gradient (10 → 30% sucrose in 20 mM Tris-HCl pH 7.5, 2.0 mM DTT, 4.0 mM MgCl₂, 0.5 M KCl). After analysis by SDS-PAGE, uncontaminated fractions were pooled, and exchanged into 20 mM Tris-HCl pH 7.5, 100 mM KCl, 2.0 mM MgCl₂, 2.0 mM DTT, 250 mM sucrose using Amicon centrifugal concentrators (4 °C, 100 K MWCO). Ribosome subunits were snap-frozen in liquid nitrogen and stored at −80 °C until required.

**Ribosome binding assays**. Assays were conducted in 50 mM Tris-acetate pH 7.5, 150 mM potassium acetate, 5.0 mM magnesium acetate, 0.25 mM spermidine, 10 mM DTT, 0.1% v/v Triton X-100. Per 60 µL binding reaction, ribosome sub-units were diluted to a final concentration of 0.4 µM, and 2A protein was added in excess to a final concentration of 2.4 µM. Twenty microlitres of this mixture was retained for SDS-PAGE analysis of the 'input'. The remaining 40 µL was incubated at room temperature for 20 min prior to application to a S200-HR size-exclusion microspin column (Cytiva) that had been pre-equilibrated (4 × 500 µL) in the above buffer by resuspension and centrifugation (300 × g, 30 s). Immediately after application, the eluate was collected by centrifugation (300 × g, 60 s).

**Western blot**. Samples were analysed by 4–20% gradient SDS-PAGE and trans-ferred to a 0.2 µm nitrocellulose membrane. All subsequent steps were carried out at room temperature. Membranes were blocked (5% w/v milk, PBS, 1 h) before incubation (1 h) with primary antibodies in 5% w/v milk, PBS, 0.1% v/v Tween-20. Membranes were washed three times with PBS, 0.1% v/v Tween-20 prior to incubation (1 h) with IRDye fluorescent antibodies in 5% w/v milk, PBS, 0.1% v/v Tween-20. After three washes in PBS, 0.1% v/v Tween-20 and a final rinse in PBS, membranes were imaged using an Odyssey CLx Imaging System (LI-COR). Figures were prepared using ImageStudio Lite 5.2 (LI-COR). Antibodies used were rabbit polyclonal anti-2A[12] (1/1000); mouse monoclonal anti-RPS6 (1/1000, clone A16009C, BioLegend); mouse monoclonal anti-RPL4 (1/1000, clone 4A3, Sigma); goat anti-rabbit IRDye 800 CW (1/10,000, LI-COR) and goat anti-mouse IRDye 680LT (1/10,000, LI-COR). Raw, uncropped blots are available in the Source Data file.

**In vitro transcription**. For in vitro frameshifting assays, we cloned a 105 nt DNA fragment (pdluc/EMCV, Supplementary Table 5) containing the EMCV slippery sequence flanked by 12 nt upstream and 86 nt downstream into the dual luciferase plasmid pDluc at the XhoI and BglII sites[80]. This sequence was inserted between the Renilla and firefly luciferase genes such that firefly luciferase expression is dependent on −1 PRF. Wild-type or mutated frameshift reporter plasmids were linearized with FspI and capped run-off transcripts generated using T7 RNA polymerase as described[81]. Messenger RNAs were recovered by phenol/chloroform extraction (1:1 v/v), desalted by centrifugation through a NucAway Spin Column (Ambion) and concentrated by ethanol precipitation. The mRNA was resuspended in water, checked for integrity by agarose gel electrophoresis, and quantified by spectrophotometry.

Messenger RNAs for 70S IC preparation (EMCV_IC, Supplementary Table 5) were produced from a 117 nt long DNA fragment containing the EMCV frameshift site flanked by the bacterial 5′ UTR with Shine-Dalgarno sequence and 18 nt downstream region of the putative structure.

5′GGGAAUUCAAAAAUUGUUAAGAAUUAAGGAGAUAUACAUA<u>AUG</u>G AGGUUUUUAUCACUCAAGGAGCGGCAGUGUCAUCAAUGGCUC**AAA** CCCUACUGCCGAACGACUUGGCCAGATCT 3′ (slippery sequence in bold, initiation codon underlined).

This sequence was PCR amplified and in vitro transcribed using T7 RNA polymerase (produced in-house). Messenger RNAs were purified using the Qiagen RNeasy midiprep kit according to the manufacturer's protocols. The mRNAs were eluted in RNAse-free water, integrity and purity was checked by gel electrophoresis and quantified by spectrophotometry.

**70S initiation complex preparation**. Ribosomes, translation factors, and tRNAs were of E. coli origin. Total E. coli tRNA was from Roche, and oligonucleotides were from Microsynth. 70S ribosomes from MRE600, EF-Tu, EF-G, IF1, IF2 and IF3 were purified from E. coli[82]. fMet-tRNA$^{fMet}$ was prepared and aminoacylated according to published protocols[83,84]. Aminoacylated fMet-tRNA$^{fMet}$ was purified by reversed-phase HPLC on a Wide Pore C5 (10 µM particle size 10 mm × 25 cm) column (Sigma Aldrich). To prepare initiation complexes, 70S ribosomes (1 µM) were incubated with a threefold excess of an EMCV model mRNA (EMCV_IC, Supplementary Table 5) encoding for 5′…AUGGA**GGUUUUU**AUC…3′ (slippery

sequence in bold) and a 1.5- fold excess each of IF1, IF2, IF3, and fMet-tRNA$^{fMet}$ in buffer A (50 mM Tris-HCl pH 7.5, 70 mM NH$_4$Cl, 30 mM KCl, 7 mM MgCl$_2$) supplemented with GTP (1 mM) for 30 min at 37 °C. 70S initiation complexes were purified by centrifugation through a 1.1 M sucrose cushion in buffer A. Before grid preparation, initiation complexes were additionally purified on Sephacryl S-300 gel filtration microspin columns.

**Frameshifting assays (in vitro translation).** Messenger RNAs were translated in nuclease-treated rabbit reticulocyte lysate (RRL) or wheat germ (WG) extracts (Promega). Typical reactions were composed of 90% v/v RRL, 20 μM amino acids (lacking methionine) and 0.2 MBq [$^{35}$S]-methionine and programmed with ~50 μg/mL template mRNA. Reactions were incubated for 1 h at 30 °C. Samples were mixed with 10 volumes of 2× Laemmli's sample buffer, boiled for 3 min and resolved by SDS-PAGE. Dried gels were exposed to a Storage Phosphor Screen (PerkinElmer) and the screen scanned in a Typhoon FLA7000 using phosphor autoradiography mode. Bands were quantified using ImageQuant™TL 8.1.0 software (GE Healthcare). The calculations of frameshifting efficiency (%FS) took into account the differential methionine content of the various products and %FS was calculated as % $-1$FS $= 100 \times$ (IFS/MetFS)/(IS/MetS $+$ IFS/MetFS). In the formula, the number of methionines in the stop and frameshift products are denoted by MetS, MetFS respectively; while the densitometry values for the same products are denoted by IS and IFS respectively. All frameshift assays were carried out a minimum of three times.

Ribosomal frameshift assays in *E. coli* employed a coupled T7/S30 in vitro translation system (Promega). A ~450 bp fragment containing the EMCV PRF signal (or mutant derivative) was prepared by PCR from plasmid pDluc/EMCV[12] and cloned into the BamHI site of the T7-based, *E. coli* expression vector pET3xc[85]. T7/S30 reaction mixes were prepared according to the manufacturer's instructions (50 μL volumes), including 10 μCi $^{35}$S methionine, supplemented with plasmid DNA (4 μg) and incubated at 37 °C for 90 min. Reactions were precipitated by the addition of an equal volume of acetone, dissolved in Laemmli's sample buffer and aliquots analysed by SDS-PAGE. PRF efficiencies were calculated as above.

**Cryo-EM specimen preparation.** Initiated 70S ribosomes in 50 mM Tris-HCl pH 7.5, 70 mM NH$_4$Cl, 30 mM KCl, 7 mM MgCl$_2$ were diluted tenfold into 20 mM HEPES pH 7.5, 100 mM potassium acetate, 1.5 mM MgCl$_2$, 2.0 mM DTT. 2 A protein was dialysed (3 K MWCO, 4 °C, 16 h) into the same buffer. Crosslinking reactions of 50 μL comprising 75 nM ribosomes, 3.0 μM 2A and 2.0 mM bis(sulfosuccinimidyl)suberate (BS3) were performed on ice (30 min) immediately prior to grid preparation. Quantifoil R 2/2 400-mesh copper supports were coated with an additional ~ 60 Å layer of amorphous, evaporated carbon by flotation[86], and thoroughly dried before use. Grids were made hydrophilic by glow-discharge in air for 30 s. Three microliters of crosslinking reaction was applied to grids which were then blotted for 4.5 s and vitrified by plunging into liquid ethane using a Vitrobot MK IV (FEI) at 4 °C, 100% relative humidity.

**Cryo-EM data collection and processing.** Micrographs were collected at the BiocEM facility (Department of Biochemistry, University of Cambridge) on a Titan Krios microscope (FEI) operating at 300 kV and equipped with a Falcon III detector (Supplementary Table 4). At ×75,000 magnification, the calibrated pixel size was 1.07 Å/pixel. Per 0.6 s acquisition in integration mode, a total exposure of 54.4 e$^-$/Å$^2$ was fractionated over 23 frames with applied defocus of $-1.5$, $-1.8$, $-2.1$, $-2.4$, $-2.7$ and $-3.0$ μm. EPU software was used for automated acquisition with five images per hole. After manual inspection, 5730 micrographs were used in subsequent image processing.

Movie frames were aligned and a dose-weighted average calculated with MotionCor 2[87]. The contrast transfer function (CTF) was estimated using CtfFind 4[88]. All subsequent image-processing steps were carried out in RELION 3.1[89] (Supplementary Fig. 6) and all reported estimates of resolution are based on the gold standard Fourier shell correlation (FSC) at 0.143, and the calculated FSC is derived from comparisons between reconstructions from two independently refined half-sets. Reference-free autopicking of 820,475 particles was performed using the Laplacian-of-Gaussian function (200–250 Å diameter). Particles were initially downscaled threefold and extracted in a 150-pixel box. Two rounds of 2D classification (into 100 and 200 classes, respectively) were used to clean the dataset to 750,029 'good' particles. An initial reference was generated from a PDB file of a 70S elongation-competent ribosome (PDB ID 5MDZ [https://doi.org/10.2210/pdb5mdz/pdb]) and low-pass filtered to 80 Å resolution. The initial 3D refinement (6.5 Å resolution) showed clear evidence for at least one copy of 2A adjacent to the factor binding site on the 30S subunit. At this stage, two rounds of focussed classification with signal subtraction were performed (6 classes) to separate particles based on additional density near (i) the factor binding site and (ii) the mRNA entry channel/helicase. The former was successful and 289,741 particles containing three copies of 2A were rescaled to full size and extracted in a 450-pixel box. Following initial 3D refinement, creation of a 15 Å low-pass filtered mask (five-pixel extension and five-pixel soft edge) and post-processing, a reconstruction of 2.93 Å was achieved. After per-particle CTF refinement and polishing, this was increased to 2.50 Å. With the increased angular accuracy provided by the fully rescaled data, focussed

classification with signal subtraction and local angular searches was performed again to separate particles based on 2A occupancy at the factor binding site. This final reconstruction (2.66 Å) from 120,749 particles revealed three copies of 2A bound with full occupancy, and clearer details in the vicinity of the 2A binding sites. Calculation of a local resolution map revealed additional low-resolution density adjacent to the beak of the 30S head. Subsequent focussed classification with signal subtraction and refinement confirmed that this was a fourth copy of 2A bound, present in 73,059 particles.

To build the model, the atomic coordinates for a 70S initiation complex (5MDZ [https://doi.org/10.2210/pdb5mdz/pdb]) and three copies of chain A from the 2A crystal structure (above) were docked as rigid bodies into the EM map. Local rebuilding was performed iteratively in COOT[63] and the models refined using phenix real-space refine[64] implementing reference model restraints to preserve geometry.

**Visualisation of structural data.** All structural figures depicting crystallographic data (cartoon, stick and surface representations) were rendered in PyMOL 2.3.4 (Schrödinger LLC). Structural figures of EM maps with docked components were rendered in ChimeraX 1.1[90].

**Reporting summary.** Further information on research design is available in the Nature Research Reporting Summary linked to this article.

## Data availability
The atomic coordinates and structure factors for the EMCV 2A X-ray crystal structure have been deposited in the wwPDB database under accession code 7BNY. The 70S IC:2A cryo-EM map has been deposited in the EMDB under accession code EMD-12635 and the refined atomic coordinates accompanying this structure have been deposited to the wwPDB under accession code 7NWT. Previously published structures that were used in this study are also available in the wwPDB: 5WE6, 4V7D and 5MDZ. Source data are provided with this paper. All raw data (e.g. uncropped, unannotated gels, western blots, tables of force measurements, MST traces) corresponding to individual figure panels are provided in the Source Data File and have also been deposited in Mendeley Data [https://doi.org/10.17632/gkpwngy65h.2]. Source data are provided with this paper.

## Code availability
The force spectroscopy analysis scripts supporting the current study have been uploaded to GitHub [https://github.com/REMI-HIRI/EMCV_2A_project]. Further information is available on request from Neva Caliskan (neva.caliskan@helmholtz-hiri.de).

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

## Acknowledgements

We thank Dima Chirgadze, Steve Hardwick and Lee Cooper (BiocEM facility) for assistance with CryoEM data acquisition. We thank Ann Mukhortava and Bärbel Lorenz (Lumicks AG) for technical assistance with optical tweezer experiments and Ann Mukhortava for the critical reading of the manuscript. We thank Prof. Marina V. Rodnina for providing expression constructs. We thank Matthias Zimmer, Trevor Sweeney, Janet Deane and Tatyana Koch for experimental assistance. We thank Vish Chandrasekaran, Jailson Brito Querido, Sebastian Kraatz and Chris Rae for helpful discussions. A Titan V graphics card used for this research was donated by the NVIDIA Corporation. Remote synchrotron access was supported in part by the EU FP7 infrastructure grant BIOSTRUCT-X (Contract No. 283570). We thank the staff of Diamond Light Source beamline I03 for assistance with crystal screening and data collection. Part of this work was carried out in the laboratory of V. Ramakrishnan, who was funded by the UK Medical Research Council (MC_U105184332), and a Wellcome Trust Senior Investigator award (WT096570). C.H.H. and S.N. were supported by a Wellcome Trust Investigator Award (202797/Z/16/Z) to I.B. C.H.H. is funded by a Sir Henry Dale fellowship (221818/Z/20/Z) from the Wellcome Trust and the Royal Society. A.E.F. is supported by Wellcome Trust (106207/Z/14/Z) and European Research Council (646891) grants to A.E.F. S.C.G. is funded by a Sir Henry Dale fellowship (098406/Z/12/B) from the Wellcome Trust and the Royal Society. N.C., L.P. and A.K. are supported by the Helmholtz Association. N.C. is funded by the European Research Council StG (948636).

## Author contributions

C.H.H. and S.N. cloned expressed and purified proteins and performed most of the biochemical experiments. L.P. designed OT constructs and performed single-molecule optical tweezers experiments and analyses under the supervision of N.C. C.H.H. and S.C.G. performed crystallography experiments. A.K. and N.C. performed MST experiments and analyses. N.C. prepared and purified bacterial initiation complexes for structural analysis. C.H.H. prepared cryo-EM grids and collected and processed cryo-EM data. C.H.H., S.N., A.F., N.C. and I.B. wrote the manuscript with contributions from all authors.

## Competing interests

The authors declare no competing interests.
