## [Peer Review File · Nature Communications]

Structural and molecular basis for Cardiovirus 2A protein as a viral gene expression switchREVIEWER COMMENTS

Reviewer #1 (Remarks to the Author):

This excellent paper by Hill et al walks through the structures of the enigmatic EMCV 2A protein, characterizing the viral RNA motifs and ribosome binding footprints that contribute to its multifunctional activities. Through adept use of crystallography, molecular tweezers and cryoEM techniques, beautifully illustrated in the abundant figure panels and supplemental materials, long-standing questions about this novel-fold protein are resolved, including a) the basic structure, relative disorder of certain loops, the incorporation of SO₄ ions, b) recognition features (structure/sequence) in the viral 2B gene PRF required for ribosomal frameshifting, and as influenced by 2A binding, c) 2A direct multi-conformational interactions with rbs RNA in intact, functional ribosomes.

The collective is well described, thorough, compelling evidence for multiple creative new host-control mechanisms used by this virus. The paper finally informs us how all these diverse elements interact to bring about a truly remarkable regulatory system. These are very important advances that will impact many fields of RNA, protein, and virology studies.

I can suggest no compelling edits that should be required here, but as a point of virology curiosity might ask that the Aus include a short comment on: if 2A binding to rbs and PRF is so easy to achieve by simple mixing, presumably cytoplasmically, during a native EMCV infection, why is 2A predominantly nucleolar? If mixed with ribosomes, it can surely bind as demonstrated here. In cells, that's not where most of it is. Might the native protein form similar or other attachments during ribosome biogenesis? Would this affect proposed hypotheses about host translation inhibition possibilities? The important Arg-rich surface loop is also functional as an effective NLS.

Reviewer #2 (Remarks to the Author):

In this manuscript, Hill et al. investigate a very interesting problem: how a novel type of frameshifting element in coronavirus that appears to be driven by the binding of a protein (protein 2A) to the viral RNA works. They combine crystallography, cryo-EM, ligand-binding assays, and single-molecule pulling experiments to characterize the structure of protein 2A, as well as its interactions with both the ribosome and the viral mRNA. They find a novel fold that allows for tight binding to the viral mRNA frameshift element, and also show that 2A can bind the rRNA at multiple sites, potentially affecting translation in other ways.

This work integrates an impressive set of different experiments to provide a thorough picture of protein 2A and its interaction with relevant RNAs. The frameshift mechanism involved here is of interest to a broad community of researchers because it is so unusual, and this manuscript provides some important insights into the features of 2A and its interactions with RNA that play essential roles in frameshifting. The manuscript is also well written.

A few issues to be addressed in revision are outlined below:

1. There is a conceptual aspect of the way the problem is described that is somewhat confusing: from the description in the manuscript, the RNA structure downstream of the slippery site does not induce any frameshifting by itself (the level is pretty close to background). But in that case, it's not really correct to call it a "stimulatory structure", since it doesn't act to stimulate frameshifting. It appears instead that this RNA is acting mainly as a binding site for 2A, and it's the action of 2A that induces frameshifting in some manner. Is there some reason to believe that this is not the case?

If it is the case, it's a rather important distinction that really should be made (and discussed) in the manuscript, because it is directly relevant to the mechanism of action of 2A, making it unclear how 2A really acts. For example, 2A binding to the pseudoknot could change it from a structure that isn't able to induce frameshifting to one that is (this explanation is the one that is favored by the authors, it would appear). But it could also be that the RNA pseudoknot has nothing to do with inducing frameshifting, instead the frameshifting is stimulated entirely by the interaction between 2A and the ribosome. If possible, the authors should attempt to devise an assay that could distinguish between these different pictures, since it relates to the central question of the mechanism of 2A in stimulating frameshifting.

2. At some locations the authors claim that the recognition of RNA by protein 2A is specific, but in other places (like line 306) they state that 2A "can recognize a diverse set of RNA targets", and the proposed mechanism of electrostatic recognition would seem to be relatively non-specific. Is the idea that it can recognize a diverse set of RNA targets that nevertheless share some specific features? Some clarification is needed. As a related point, if 2A can recognize a diverse set of RNA targets, does it also bind to other RNA pseudoknots of similar size?

3. Some issues regarding the optical tweezers measurements:

(a) Please indicate the number of molecules per construct/condition as well as the number of pulls.

(b) Please clarify if the results for the length changes are meant to represent end-to-end extension change or contour length change. The length changes should be expressed in terms of contour length change, which is force-independent. In contrast, the extension is a force-dependent variable.

(c) Please include in the table of results the lengths that would be expected for the proposed structures, so that readers can assess if the proposals are reasonable.

(d) The description of the structural sub-populations is somewhat unclear: for example, when the authors state (line 180) that 42% of molecules were in state 3, do they mean that 42% of the molecules showed this state 100% of the time (the literal meaning of what is written), or that 42% of the pulling curves showed this state (and the fraction was ~42% for all molecules)? If it is the former, there is something strange going on, since that would mean that the RNA molecules are actually different in some way. Please clarify, and explain if the molecules did not all show roughly the same behavior.

(e) With 4 sub-populations present, the number of pulling curves (60–160 per condition) is not really enough to quantify the occupancies of all the sub-populations reliably, especially for the low-occupancy states. This uncertainty should be acknowledged and taken into account in the comparisons for 2A present/absent. It doesn't look like a big problem, fortunately, given the size of the occupancy change with 2A present.

(f) Was an oxygen scavenger system used to prevent damage to the RNA from oxygen radicals generated by photodissociation? Such damage is a known problem (see for example Landry et al. (2009), *Biophys. J.* 97:2128). Given the relatively small-number statistics involved in some of the sub-populations, it's important to know if they might be artifacts from oxidative damage.

(g) One aspect of the interpretation of the tweezers data is not very solid: for the sub-population of behavior that is listed as "state 5", no rips are seen. But then how can the authors be sure that these curves represent a single population? Suppose that 2A can bind to state 4 *and* to state 3, and in each case it stabilizes the structure so that it unfolds at over 40 pN. How would these two states be distinguished? The contour length difference between the two would be sufficiently small that it is doubtful that they could be distinguished that way. The authors should discuss this possibility, and any evidence for/against it.

Minor points:

(i) The authors claim that the fold of 2A is novel. How is it determined that there is no homology to other protein folds?

(ii) On line 344, please clarify if the interactions seen in the cryo-EM images are with rRNA or mRNA (or both).

(iii) Please define acronyms like TMEV and NLS.

(iv) In Fig. 3, please include a zoom into the rip region in panel 3b, the rips are very small and hard to see as shown. Also, the labels 1-2-3 in Fig. 3b are quite confusing—presumably they are meant to be ordinal numbers (1st, 2nd, 3rd), but they might also refer to state 1, state 2, and state 3. Please use a less

ambiguous way of showing this. It is also unclear what the x-axis label ("Rel. distance") refers to in these panels. Presumably this is the distance between the beads; if so, then it is standard to refer to it as the extension (more precisely, end-to-end extension). Finally, the "heat maps" in Fig. 3c and 3d are a very odd way of presenting the data, please convert to the standard in the field, which is to present force distributions as histograms (which show the counts in each bin) or probability densities (area normalized to 1).

(v) The Introduction (lines 42-43) states that "the stability and unfolding kinetics of these stimulatory elements are thought to be the primary determinants of PRF efficiency," but the Discussion (lines 366-368) makes it clear that the situation is more complex. It would be helpful to revise the statement in the introduction so that it includes the appropriate nuance.

(vi) Mechanical stabilization of nucleic acid structures by protein binding is a well known phenomenon. A citation of either some relevant primary literature or else a review such as Heller et al. (2014) *Chem. Rev.* 114:3087) would be appropriate.

Reviewer #3 (Remarks to the Author):

Mechanisms of programmed frame-shifting (PRF) during translation induced by structural elements (hairpins, pseudoknots) downstream of the 'slippery sequence' at which PRF occurs have been extensively characterized. In the present manuscript, Hill et al. have used a variety of complementary approaches, including X-ray crystallography, single-molecule optical tweezer-based analysis of RNA unwinding, cryo-electron microscopy and functional assays to characterize an unusually divergent mechanism of frameshifting used by cardioviruses (in this instance, encephalomyocarditis virus (EMCV)) that depends on binding of a virus-encoded protein ("2A") to a structural element in the genome-length mRNA. The authors report the crystal structure of EMCV 2A, determining that it has a novel fold (a beta sheet backed by two alpha helices) with an extensive positively charged solvent-exposed surface and loops. 2A binds the cognate frame-shift stimulatory sequence with high affinity, and interestingly, analysis of the minimal binding element led the authors to revise their model for it, such that it appears to consist of a pseudoknot rather than just a hairpin. This revised structural model was supported by the results of mutational analysis. Single-molecule measurements of unwinding of this element in the presence of 2A suggested that it stabilizes this element (i.e. increases its resistance to unwinding) and could thus contribute to PRF by imposing a delay on elongation.

2A is known to bind to the mammalian small (40S) ribosomal subunit, and Hill et al. confirmed this and determined that it also binds with high affinity to the equivalent bacterial 30S subunits. 2A-dependent PRF can be recapitulated in bacterial translation extracts, and the authors use this observation to justify determination of the structure of bacterial ribosomes bound to 2A by cryo-electron microscopy. This element of the manuscript is the least conclusive. The authors observed three (and in a subset of complexes, four) molecules of 2A bound to the 30S subunit, yielding insights into how 2A recognizes specific RNA targets. However, they do not comment on whether binding sites are conserved in eukaryotic 40S ribosomal subunits (which would be informative and should be done). They also do not comment on whether the stoichiometry and affinity of binding of 2A to eukaryotic and bacterial ribosomal is comparable, so that the direct relevance of these structural observations to PRF in the natural host remains unknown. Although the location of 2A binding sites suggest that 2A might interfere with binding of elongation factors (and thus lead to a kinetic delay in elongation that might enhance PRF), the authors were not able to distinguish between the relative importance of 2A's binding to the viral mRNA and to ribosomal subunits in mediating PRF. They note that binding of 2A to the ribosome and to viral mRNA involves the same elements of 2A and would thus be mutually exclusive. Indeed, saturation of ribosomes with 2A late in infection might contribute more to a shut-off of translation than to PRF.

In conclusion, this manuscript reports several important advances regarding a mechanism of protein-dependent PRF that the authors have established as a paradigm for this type of process. The results are significant, clearly presented and likely to be of general interest.

Specific comment.

1. Lines 822-823. The description of the method used for purification of ribosomal subunits from rabbit reticulocytes includes an incorrect reference. This cited paper refers to two relevant papers, both of which concern E. coli ribosomes rather than mammalian ribosomes.

Reviewer #4 (Remarks to the Author):

The manuscript by Hill et al. addresses the mechanism underlying the function of a viral protein, 2A. The work includes the crystal structure determination of 2A, interaction analysis with mRNA regions relevant for programmed -1 ribosomal frameshifting, complex formation with the small ribosomal subunit and cryo-EM structure determination of a bacterial 70S translation initiation complex with 2A bound, complemented with single molecule analysis. This impressive integrated study hence provides a number of novel findings, which are overall clearly presented. Nevertheless, some parts would benefit from a

clearer discussion. For example, it is not clear why there should be multiple binding sites for 2A on the ribosome (is that not just an effect of using a large excess of 2A for the biochemical reconstitution of the complex?), while the primary target described as being selective is the viral mRNA. Also, as 2A targets eukaryotes, how relevant is it to use a bacterial ribosome to reconstitute a complex with 2A? That frameshifting works in both species doesn't explain this per se, the binding sites could still be different. What is the meaning of the covalently bound dimer of 2A in the context of mRNA recognition or presence on the ribosome, there seem to be no dimers involved there; this could be discussed.

Detailed points:

- thank you for sharing maps and atomic models. The crystal structure looks fine; minor point: what is the meaning of the Arg109 –SO4²⁻ ion? Sounds non-specific. Cryo-EM map: the refinement of the atomic model looks good. However, the contour level needs to be adjusted rather strongly between the different 2A molecules. This is a sign that the stoichiometry is rather variable, possibly answering the question of functional relevance of these, i.e. in fact there might be only one really relevant. It is surprising to see several 2A molecules anyway, is the RNA fold recognition similar? A super-position may help (see point below). The density is good for one, but hardly resolves the secondary structure of the others. This should be clarified in the text to not give the impression of 3 (or 4?) well defined molecules; again, this suggest that one is well bound. What is the excess used to prepare the complex? This should be discussed in this context. Finally, there is a large two-domain density between the 30S beak and the 50S ribosomal subunit, which is not accounted for – yet another molecule?

- crystal structure of 2A: MALS seems to indicate a monomer, but the structure shows a covalently bound dimer. Are there any such dimers on the viral mRNA or on the ribosome (it doesn't seem so)?

- as there are 4 molecules in the ASU with NCS, what are the structural differences (only briefly mentioned) and what could these tell for 2A function?

- line 130: binding of 2A to EMCV: should probably say EMCV mRNA

- pages 6 and 7 on single molecule measurements: is a relatively lengthy section with a number of hypotheses but not that conclusive, could be more concise

- line 265: the hypothesis that 50S ribosomal subunit joining and 70S IC formation is compatible with the presence of 2A should be carefully analysed. What would it mean that 2A remains bound in a 70S IC complex? When would it dissociate? e.g. in a study a few years ago among the initiation factors IF1, IF2 and IF3, IF3 had been proposed to remain bound, but that turned out later to be incorrect (an rRNA segment had been interpreted as IF3). In this context it is also not clear what the multiple binding sites of 2A mean, are they all relevant?

- line 273: how do the 7% or 15% (on a bacteria-optimized mRNA linker length) relate to the amount of frameshifted protein needed / produced in cellulo by the virus?

- what is the meaning of 3 or 4 2A copies bound on the ribosome? Are these significant? A structural comparison / superposition might help. Are the binding regions conserved between prokaryotic and eukaryotic ribosomes? If not, what is the meaning to have done the structural analysis on a bacterial ribosome as the primary target of the virus is not bacterial apparently?

- line 381: EF-G “catalysed”: as translocation is not an enzymatic step by itself it would be better to call this “mediated”; in fact, translocation can happen even without EF-G, but at slower rates
- legend Fig. 5: the term is “focused” classification and refinement, e.g. see references such as Curr Op Struct Biol 2017, eLife 2018, Biosci Rep 2018 etc.; to be corrected throughout, e.g. Suppl. Fig. 6 etc.
- legend Fig. 5: the term cryo-EM “electron” density is technically incorrect. Due to the nature of electrons, which are charged, these are not electron density maps as in X-ray crystallography. Instead, these are electrostatic potential maps (see e.g. Wang & Moore, 2017; Hryc et al., 2017; Marques et al., 2019; Wang et al., 2021). A common way of saying would be simply “cryo-EM map”.
- Fig. 6: is the stem-loop structure a hypothesis / structure prediction from mRNA fold prediction? If so, it should be mentioned, especially as the pseudoknot structure now comes from this study
- it is good that the structures and maps will be deposited. For Table 1, Rmerge is an outdated value and can/should be removed
- line 949: instead of “initial model” a better term would be “initial reference”
- line 958: CTF-refinement gave 2.5 Å resolution, but the final reconstruction is at 2.66 Å, why is that? Which map provides more details or less noise?

Reviewer #1 (Remarks to the Author):

“This excellent paper by Hill et al walks through the structures of the enigmatic EMCV 2A protein, characterizing the viral RNA motifs and ribosome binding footprints that contribute to its multifunctional activities. Through adept use of crystallography, molecular tweezers and cryoEM techniques, beautifully illustrated in the abundant figure panels and supplemental materials, long-standing questions about this novel-fold protein are resolved, including a) the basic structure, relative disorder of certain loops, the incorporation of SO₄ ions, b) recognition features (structure/sequence) in the viral 2B gene PRF required for ribosomal frameshifting, and as influenced by 2A binding, c) 2A direct multi-conformational interactions with rbs RNA in intact, functional ribosomes.

The collective is well described, thorough, compelling evidence for multiple creative new host-control mechanisms used by this virus. The paper finally informs us how all these diverse elements interact to bring about a truly remarkable regulatory system. These are very important advances that will impact many fields of RNA, protein, and virology studies.”

We thank the reviewer for their kind remarks and enthusiasm for the subject matter.

“I can suggest no compelling edits that should be required here, but as a point of virology curiosity might ask that the Aus include a short comment on: if 2A binding to rbs and PRF is so easy to achieve by simple mixing, presumably cytoplasmically, during an native EMCV infection, why is 2A predominantly nucleolar? If mixed with ribosomes, it can surely bind as demonstrated here. In cells, that’s not where most of it is. Might the native protein form similar or other attachments during ribosome biogenesis? Would this affect proposed hypotheses about host translation inhibition possibilities? The important Arg-rich surface loop is also functional as an effective NLS.”

Stimulation of frameshifting must utilise a cytoplasmic pool of 2A, as this is where viral translation takes place. However, the reviewer raises a very interesting point – indeed, the “arginine loop” acts as a functional NLS and 2A staining in infected cells is nucleolar. We have refrained from speculation in the text, but we note that earlier work demonstrating co-fractionation of 2A with 40S in infected cells (PMID: 17728235) identified that a small proportion of 2A remained tightly bound to 40S despite high-salt washes (750 mM). Given the potential ability of 2A to bind to multiple sites, it is an intriguing possibility that nucleolar 2A may also be able to bind to immature ribosomal RNA during biogenesis. The existence of several populations of 2A-40S with different salt sensitivity (PMID: 17728235) implies that there may be several modes of interaction. The functional consequences of this on host vs. viral translation would depend on how and where 2A was incorporated, as well as 2A concentration during infection. Future work will investigate this in mammalian cells.

Reviewer #2 (Remarks to the Author):

“In this manuscript, Hill et al. investigate a very interesting problem: how a novel type of frameshifting element in cardiovirus that appears to be driven by the binding of a protein (protein 2A) to the viral RNA works. They combine crystallography, cryo-EM, ligand-binding assays, and single-molecule pulling experiments to characterize the structure of protein 2A, as well as its interactions with both the ribosome and the viral mRNA. They find a novel fold that allows for tight binding to the viral mRNA frameshift element, and also show that 2A can bind the rRNA at multiple sites, potentially affecting translation in other ways.

This work integrates an impressive set of different experiments to provide a thorough picture of protein 2A and its interaction with relevant RNAs. The frameshift mechanism involved here is of interest to a broad community of researchers because it is so unusual, and this manuscript provides some important insights into the features of 2A and its interactions with RNA that play essential roles in frameshifting. The manuscript is also well written.”

We thank the reviewer for highlighting the novelty and general interest of our findings, and for the kind remark about the quality of writing.

“A few issues to be addressed in revision are outlined below:

1. There is a conceptual aspect of the way the problem is described that is somewhat confusing: from the description in the manuscript, the RNA structure downstream of the slippery site does not induce any frameshifting by itself (the level is pretty close to background). But in that case, it’s not really correct to call it a “stimulatory structure”, since it doesn’t act to stimulate frameshifting. It appears instead that this RNA is acting mainly as a binding site for 2A, and it’s the action of 2A that induces frameshifting in some manner. Is there some reason to believe that this is not the case?”

We apologise for any confusion – we describe the structured RNA element as “stimulatory” in line with conventions in the field of protein-independent PRF. More precisely, this structured RNA element is necessary but not sufficient for -1 PRF at a permissive slippery sequence. We have now clarified this in the introduction.

“If it is the case, it’s a rather important distinction that really should be made (and discussed) in the manuscript, because it is directly relevant to the mechanism of action of 2A, making it unclear how 2A really acts. For example, 2A binding to the pseudoknot could change it from a structure that isn’t able to induce frameshifting to one that is (this explanation is the one that is favored by the authors, it would appear). But it could also be that the RNA pseudoknot has nothing to do with inducing frameshifting, instead the frameshifting is stimulated entirely by the interaction between 2A and the ribosome. If possible, the authors should attempt to devise an assay that could distinguish between these different pictures, since it relates to the central question of the mechanism of 2A in stimulating frameshifting.”

Here we present strong evidence that the primary determinant of -1 PRF is 2A binding to specific conformations of the stimulatory element in the vRNA. In **Figure S3b and c**, we show that even single-nucleotide mutations in the RNA stimulatory element sequence (G7C or C37G) disable -1 PRF *in vitro* by preventing 2A binding to the stimulatory element. Both of these individual mutations are predicted to prevent formation of the pseudoknot-like conformation. However, the double mutation (G7C+C37G) that would restore this conformation reinstates both 2A binding (**Figure S3c**) and frameshifting (**Figure S3b**). In all of these assays, nothing was done to prevent 2A interacting with the ribosome, so the loss of -1 PRF is most likely to result in impaired 2A interaction with the stimulatory RNA element.

We have also previously demonstrated that mutation of the conserved cytosine triplet in the stimulatory element (CCC  CUC; PMID: 28593994) prevents -1 PRF by inhibiting 2A binding. Consistent with our above biochemical observations, our optical tweezers measurements show that the distribution of conformers with CUC RNA is depleted for pseudoknot-like states relative to wild-type, and furthermore, this distribution is unresponsive to addition of 2A protein (**Figure 3e and S4e, f**).

Nevertheless, the reviewer raises an important point – all of the above observations do not exclude the possibility of a secondary interaction with the ribosome whilst 2A remains bound at the RNA stimulatory element. This alternative model would necessitate two molecular surfaces on 2A: one for binding to the stimulatory element and one for binding to the ribosome. To test this, we prepared a series of 2A mutants based on the ribosome interaction surface observed in our 70S-2A cryo-EM structure. In **Figure S5** we show that 2A uses this same molecular surface (comprising R95, R97, K73, K50, K48, R46) to bind the stimulatory element. This implies that, for each 2A molecule, these binding events are mutually exclusive. Additionally, the severity of the -1 PRF defect correlates with the ability of the mutant 2A to bind to the stimulatory element (see **Figure S5g and h**). Taken together, this indicates that a specific interaction between 2A and the ribosome is likely not the primary determinant of -1 PRF efficiency. We have now clarified these points in the discussion.

“2. At some locations the authors claim that the recognition of RNA by protein 2A is specific, but in other places (like line 306) they state that 2A “can recognize a diverse set of RNA targets”, and the proposed mechanism of electrostatic recognition would seem to be relatively non-specific. Is the idea that it can recognize a diverse set of RNA targets that nevertheless share some specific features? Some clarification is needed.”

We apologise that our explanation in the text wasn't clear. The electrostatic nature of the interaction was inferred from the interface observed in the 70S-2A cryo-EM structure and confirmed by mutagenesis. However, the three copies of 2A observed in this structure are recognising different RNA backbone conformations (see **Figure 5e-g**, insets). Stem-loops (e.g. **Figure 2c and d**) are not recognised by 2A.

To better illustrate this point, we include some additional analyses (**Response Figure R1 and new Supplementary Figure 7**). We first defined the overall surface patch on the 16S rRNA, based on < 8.0 Å distance to any of the three 2A molecules in the complex structure with 70S IC (**Figure R1a**). Following a PDBePISA analysis of molecular contacts, we then extracted the RNA binding sites corresponding to individual 2A molecules (**Figure R1b**). Pairwise backbone alignment of these sites by least-square superposition shows that they are considerably variable (**Figure R1c**), however they all feature 3-4 segments of phosphodiester backbone, often bridging two or more rRNA helices. An alternative way of comparing the sites is to superimpose them based on first aligning the three 2A molecules (**Figure R1d and R1e**). This approach also fails to identify a structurally conserved conformational backbone motif, illustrating how 2A may be able to recognise multiple RNA targets.

The best explanation of the exact 'rules' governing 2A specificity would come from the structure of the complex between 2A and its primary target - the RNA stimulatory element. Efforts to elucidate this are ongoing, but all our observations to date are consistent with there being a conformational requirement of the phosphodiester backbone, given that i) 2A does not bind to stem-loops and ii) we do not observe non-specific 2A binding all over the ribosome (e.g. to any of the many exposed RNA helices on the large 50S subunit).

Also see response to reviewer three and four below – briefly, the 2A1 binding site is the most conserved between 30S and 40S, as well as having the best cryo-EM density in the map, therefore this is likely to be the highest affinity site and the most physiologically relevant. It is possible that 2A2 and 2A3 are lower affinity sites which we observed due to the large molar excess used for complex preparation. We have changed the results and discussion to clarify this.

Response Figure R1 – Superposition of the three 2A rRNA binding sites in the 70S IC: 2A complex structure fails to reveal a common recognition motif. **a**, diagram showing the binding sites of each 2A molecule (coloured sticks) on the 16S rRNA (grey cartoon). rRNA residues involved in each molecular interface were determined by PDBePISA. **b**, as above, showing each of the 2A binding sites separately (sticks), superimposed on local 16S rRNA (grey cartoon). **c**, backbone superposition of the three sites fails to reveal a common binding motif. **d**, superposition of binding sites (sticks) based on alignment of the three 2A molecules (grey ribbon diagrams). Two orthogonal views are shown. *<Inset>* cartoon representation of RNA backbone shown in the top-down view. **e**, as in d, but trimmed to show spatial conservation (i.e. only residues within 3.0 Å of those in other binding sites).

“As a related point, if 2A can recognize a diverse set of RNA targets, does it also bind to other RNA pseudoknots of similar size?”

We thank the reviewer for this suggestion. To test this, we conducted EMSA experiments to compare the ability of 2A to bind the minimal 47 nucleotide EMCV stimulatory element and an unrelated 48 nucleotide two-stem pseudoknot from Infectious bronchitis virus (IBV) (**Figure R2**). No binding to the IBV RNA is observed, even at concentrations of $\sim 30 \mu\text{M}$ 2A, reinforcing the idea that this protein exhibits strong selectivity for the EMCV-specific RNA conformation. The precise mechanism of molecular recognition between 2A and its cognate RNA element is the subject of ongoing structural work.

Response Figure R2 – EMSA analyses of 2A binding to the EMCV minimal RNA stimulatory element (EMCV6, upper) and an unrelated pseudoknot from IBV of similar size (lower). RNA was 5' labelled with ^{32}P and used at a final concentration of 10 nM. The experiment was conducted twice as described in **Methods**, and representative gels are shown.

“3. Some issues regarding the optical tweezers measurements:

(a) Please indicate the number of molecules per construct/condition as well as the number of pulls.”

We thank the reviewer for their suggestion. We have included the number of individual measurements and number of tethers in main text **Figure 3e and 3f** and **Supplementary Table 3**. Briefly, a minimum of 85 individual curves were recorded on about 15-20 tethers per RNA sample.

“(b) Please clarify if the results for the length changes are meant to represent end-to-end extension change of contour length change. The length changes should be expressed in terms of contour length change, which is force-independent. In contrast, the extension is a force-dependent variable.”

We thank the reviewer for their careful attention to the data. Their comments prompted us to do a thorough review of how we were fitting and analysing the data. Specifically, we have now performed mathematical fits for all force-distance trajectories (**Methods**) in order to calculate the contour-length change, which allowed us to efficiently identify artefacts caused by experimental noise, multiple tethers or incomplete hybridization of the RNA:DNA handles, and not include them in the analyses.

An important point to note is that the predicted nucleotide length for the putative PK and an extended stem loop are similar, therefore these two conformers cannot be easily distinguished from the change in contour length. Thus, to better discriminate conformational differences of the RNA element, we now also included work performed during folding and unfolding in order to calculate the free energies (**Supplementary Table 3**). These results match very well with the *mfold* predictions of the putative conformations.

“(c) Please include in the table of results the lengths that would be expected for the proposed structures, so that readers can assess if the proposals are reasonable.”

We have now included the 2D representations of the stem loop, extended stem loop predicted by *mfold* and also the putative pseudoknot in **Figure 3b** and also refer to them in **Figure S3** legends and **Table S6**.

“(d) The description of the structural sub-populations is somewhat unclear: for example, when the authors state (line 180) that 42% of molecules were in state 3, do they mean that 42% of the molecules showed this state 100% of the time (the literal meaning of what is written), or that 42% of the pulling curves showed this state (and the fraction was ~42% for all molecules)? If it is the former, there is something strange going on, since that would mean that the RNA molecules are actually different in some way. Please clarify, and explain if the molecules did not all show roughly the same behavior.”

We apologize for the confusion. In the text, 42% referred to individual pulling curves. We have changed the wording in the text to make this point clear.

“(e) With 4 sub-populations present, the number of pulling curves (60–160 per condition) is not really enough to quantify the occupancies of all the sub-populations reliably, especially for the low-occupancy states. This uncertainty should be acknowledged and taken into account in the comparisons for 2A present/absent. It doesn’t look like a big problem, fortunately, given the size of the occupancy change with 2A present.”

We acknowledge the reviewer’s valuable comments. Based on the mathematical fitting on all (un)refolding trajectories we were able to better evaluate the low occupancy populations and states with no visible unfolding rip. Previously, the sub-populations were sorted based on the change in their force distribution. Our new data analysis based on contour lengths, revealed that some of the states are almost identical, but showing differences in the force of unfolding. We were also able to eliminate technical outliers, which could be due to mis-annealing or partial multi-tether formation. In addition, to clarify whether the previous State 5 with no visible rip is the stabilized state, we have conducted more measurements where we increased the force until a rip can be observed during measurements, meaning beyond 35-40 pN.

We still acknowledge the presence of different conformers, yet given the sample size, to avoid any doubts on state assignments, we refrain from using states or sorting the data based on force distributions. Instead, we provide a global view of the data by plotting histograms for force, contour length distributions in the presence and absence of 2A, which clearly represents the change in physical properties of the RNA in the presence of 2A (**Figure S4**). The results and discussion have now been rewritten based on these points. We hope that these substantially improved the clarity of the results and the narrative of the text.

We would like to emphasize that achieving a larger sample size in the presence of 2A protein was very challenging, since the protein is very prone to aggregation during measurements. Overall, based on the significant difference in the occupancy of >20 pN unfolding events in the presence of 2A (**Figure 3e**), the main conclusion of the OT experiments “EMCV RNA structure is stabilised in the presence of 2A” is still valid. Our single molecule data also confirms the biochemically suggested structural transition of the EMCV stem loop in the presence of 2A. In addition, lack of this transition in CUC mutant RNA strongly supports that the effect of 2A on the RNA is specific, and 2A mainly alters the unfolding rather than the refolding of the EMCV RNA (**Figure 3f**).

“(f) Was an oxygen scavenger system used to prevent damage to the RNA from oxygen radicals generated by photodissociation? Such damage is a known problem (see for example Landry et al. (2009), Biophys. J. 97:2128). Given the relatively small-number statistics involved in some of the sub-populations, it's important to know if they might be artifacts from oxidative damage.”

We thank the reviewer for raising the possibility of artefacts caused by oxygen induced RNA/tether damage. We cannot rule out that some stochastic noise exists in the system during OT measurements, which could be due to oxygen radicals, but could have many other reasons, such as aggregation of 2A, incomplete hybridization of the tether, or non-specific interactions between RNA and protein.

In OT measurements we employ oxygen scavengers only during fluorescence measurements, while no significant difference was noted in our early force-ramp experiments in the presence and absence of scavengers. In addition, although benefits of oxygen scavengers are clear from the study above, some oxygen scavengers like glucose oxidase and catalase were reported to bring unavoidable pH decrease over time, which can also affect the structure and stability of single stranded nucleic acids (PMID: 29285923). The recommendation from the technical specialists in Lumicks was to reduce the amount of the scavengers or that they could not be added at all. It was also suggested to use oxygen scavengers which do not change the pH (PMID: 22703450). We speculate using microfluidics in the Lumicks C-TRAP system also reduces the chance of additional oxygen entering the system (100% sealed system), and the constant flow of freshly degassed buffer also reduces the concentration of singlet oxygen. We added also DTT as a reducing agent in our buffers, which may reduce the ROS to some extent.

To see if the lack of oxygen scavengers affected our tether stability, we compared the measurements performed on the same tether/RNA molecule over time. Overall unfolding and refolding traces in our wild type and CUC experiments showed reversible behaviour at early and late measurements of the tether (**Figure R3**). Also, according to the Landry et al., the most affected part of the bead-tether-bead would be the AD-antibody linkages. So, the bigger the bead is, the further away is this linkage from the IR laser, and hence tether stability would be expected to increase. In our experiments, using the 6X larger sized AD beads (2.12 μm) might have partially helped to avoid the laser induced damage.

We feel that our choice of beads and mobile phase represents a good compromise between low generation of oxygen radicals and stable experimental pH. We have added a text confirming that oxygen scavengers were not used, and our justification of this choice, to the Methods. Nevertheless, for future work, we'll consider adding scavengers, especially those that are not altering pH like pyranose oxidase and catalase, particularly if we study RNA conformations alone to better examine the fine differences in the structure.

Response Figure R3. Comparison of force-distance trajectories for an individual tether at 1,3,4, 10 and 14th (left to right) pull cycle. The experiment was conducted with the EMCV WT RNA sample.

*“(g) One aspect of the interpretation of the tweezers data is not very solid: for the sub-population of behavior that is listed as “state 5”, no rips are seen. But then how can the authors be sure that these curves represent a single population? Suppose that 2A can bind to state 4 *and* to state 3, and in each case it stabilizes the structure so that it unfolds at over 40 pN. How would these two states be distinguished? The contour length difference between the two would be sufficiently small that it is doubtful that they could be distinguished that way. The authors should discuss this possibility, and any evidence for/against it.”*

To better understand the sub-populations and absence of rip in some traces, we now performed additional measurements at higher forces until we observe a rip. Previously data sorting was done based on force distributions as mentioned in point (e), now we added contour length as a constraint to re-evaluate the (un)folding curves and overcome potential technical artefacts due to multi-tether formation and hybridization issues.

Nevertheless, the contour length of the stem loop and pseudoknot are very similar, (20-25 nm). Based on these fit values, we cannot confidently say whether the 2A stabilizes the pseudoknot or just the core stem-loop of EMCV PRF stimulatory RNA structure, as the contour length change could represent both (**Figure S4**). Given this overall similarity, it is possible that the 2A protein recognizes a binding motif shared by both conformations. Currently, we don't have structural data of the RNA element itself or 2A-RNA complexes, but efforts to better understand the structure are ongoing. However, we have supporting biochemical evidence in **Figure S3b** and **S3c**, where we show that even single-nucleotide mutations of the RNA element (G7C and C37G) abrogate 2A binding and –1PRF. These mutations were predicted to prevent the formation of the pseudoknot-like conformation. More strikingly, the complementary mutations to G7C+C37G was able to restore the interaction and –1PRF. These findings further support that the pseudoknot formation is favoured by 2A binding and this is possibly the driver of the stabilization effect observed. This point is clarified in the text.

“Minor points:

(i) The authors claim that the fold of 2A is novel. How is it determined that there is no homology to other protein folds?”

We apologise that this wasn't clear, we have added a sentence to the results. Prior to designation of the 'beta shell' as a new fold, structure-based database searches for proteins with similar folds to EMCV 2A were performed using PDBeFOLD, DALI and CATHEDRAL. No significant hits were obtained for any of these searches.

“(ii) On line 344, please clarify if the interactions seen in the cryo-EM images are with rRNA or mRNA (or both).”

The interactions seen cryo-EM are all between 2A and rRNA. The text has been updated to clarify this point.

“(iii) Please define acronyms like TMEV and NLS.”

Done.

“(iv) In Fig. 3, please include a zoom into the rip region in panel 3b, the rips are very small and hard to see as shown. Also, the labels 1-2-3 in Fig. 3b are quite confusing—presumably they are meant to be ordinal numbers (1st, 2nd, 3rd), but they might also refer to state 1, state 2, and state 3. Please use a less ambiguous way of showing this. It is also unclear what the x-axis label (“Rel. distance”) refers to in these panels. Presumably this is the distance between the beads; if so, then it is standard to refer to it as the extension (more precisely, end-to-end extension). Finally, the “heat maps” in Fig. 3c and 3d are a very odd way of presenting the data, please convert to the standard in the field, which is to present force distributions as histograms (which show the counts in each bin) or probability densities (area normalized to 1).”

We thank the reviewer for their valuable suggestions. We have changed the complete panel to better represent the data, included a zoom into the rip, x-axis labels are changed to “Extension, nm”. Instead of heat maps we now plot force distribution with all the rip positions represented as individual points in main text and histograms of force and contour length distributions, fitted with gaussians in the supplement (**Figure S4**).

“(v) The Introduction (lines 42-43) states that “the stability and unfolding kinetics of these stimulatory elements are thought to be the primary determinants of PRF efficiency,” but the Discussion (lines 366-368) makes it clear that the situation is more complex. It would be helpful to revise the statement in the introduction so that it includes the appropriate nuance.”

We have changed the introduction text to better represent the complexity of the kinetic and thermodynamic determinants of the process.

“(vi) Mechanical stabilization of nucleic acid structures by protein binding is a well known phenomenon. A citation of either some relevant primary literature or else a review such as Heller et al. (2014) Chem. Rev. 114:3087) would be appropriate.”

Thanks for the suggestion, we have now included this citation in the results.

Reviewer #3 (Remarks to the Author):

“Mechanisms of programmed frame-shifting (PRF) during translation induced by structural elements (hairpins, pseudoknots) downstream of the 'slippery sequence' at which PRF occurs have been extensively characterized. In the present manuscript, Hill et al. have used a variety of complementary approaches, including X-ray crystallography, single-molecule optical tweezer-based analysis of RNA unwinding, cryo-electron microscopy and functional assays to characterize an unusually divergent mechanism of frameshifting used by cardioviruses (in this instance, encephalomyocarditis virus (EMCV)) that depends on binding of a virus-encoded protein ("2A") to a structural element in the genome-length mRNA. The authors report the crystal structure of EMCV 2A, determining that it has a novel fold (a beta sheet backed by two alpha helices) with an extensive positively charged solvent-exposed surface and loops. 2A binds the cognate frame-shift stimulatory sequence with high affinity, and interestingly, analysis of the minimal binding element led the authors to revise their model for it, such that it appears to consist of a pseudoknot rather than just a hairpin. This revised structural model was supported by the results of mutational analysis. Single-molecule measurements of unwinding of this element in the presence of 2A suggested that it stabilizes this element (i.e. increases its resistance to unwinding) and could thus contribute to PRF by imposing a delay on elongation.

2A is known to bind to the mammalian small (40S) ribosomal subunit, and Hill et al. confirmed this and determined that it also binds with high affinity to the equivalent bacterial 30S subunits. 2A-dependent PRF can be recapitulated in bacterial translation extracts, and the authors use this observation to justify determination of the structure of bacterial ribosomes bound to 2A by cryo-electron microscopy. This element of the manuscript is the least conclusive. The authors observed three (and in a subset of complexes, four) molecules of 2A bound to the 30S subunit, yielding insights into how 2A recognizes specific RNA targets.”

“However, they do not comment on whether binding sites are conserved in eukaryotic 40S ribosomal subunits (which would be informative and should be done).”

We thank the reviewer for this good suggestion. **Response Figure R4** below shows this alignment and is now included as part of **Supplementary Figure 7**.

Response Figure R4 – structural superposition of 2A binding sites observed on *E. coli* 16S rRNA (30S) with equivalent sites on *O. cuniculus* 18S rRNA (40S). **a**, alignment of the bacterial 30S (cyan) and mammalian 40S (grey) subunits. The 30S 2A binding patch and equivalent 40S rRNA (black) are highlighted. *<Inset>* Close-up view of 2A binding sites after backbone superposition. **b**, Local alignment of each individual 2A binding site (coloured sticks and surface as defined in **Response Figure R1**) to nearest equivalent in 40S rRNA (black). Per site, deviation from the starting position in the 30S rRNA (green) was permitted.

Alignment of *E. coli* 30S and mammalian 40S shows that the rRNA at the 2A-binding surface is structurally similar (**Figure R4a**). Following this initial backbone superposition, local structural alignment of each individual binding site to equivalent 40S rRNA indicates that 2A1 site shows the greatest structural conservation (**Figure R4b**). The 2A2 site is partially conserved (right hand segment in **Figure R4b**), but bacterial and mammalian rRNA are conformationally divergent at the 2A3 site. Therefore, we predict that 2A1 is the most physiologically relevant binding site. This is consistent with it likely being the highest affinity site, as evidenced by it having the best cryo-EM density compared to 2A2 and 2A3 (also see comments by reviewer 4). We have now stated this in the results and discussion.

While our structural study has shown that the 2A molecule has conformational plasticity, allowing to it bind a range of RNA conformations (see **Figure 5K**), it confirms that the interactions with highest apparent affinity share common structural arrangements (see **Response Figure R4** above) and are thus the most likely to confer the 2A biological effect.

“They also do not comment on whether the stoichiometry and affinity of binding of 2A to eukaryotic and bacterial ribosomal is is comparable, so that the direct relevance of these structural observations to PRF in the natural host remains unknown.”

In **Figure 4**, our MST experiments demonstrate a consistent preference for small subunit binding to both bacterial and eukaryotic ribosomes, with comparable apparent K_D values of ~10 and ~4 nM, respectively. We agree with the reviewer that this does not unequivocally suggest that they are

binding at equivalent sites. Due to technical issues with protein aggregation, it was not possible to obtain reliable stoichiometric information from our MST titrations. Therefore, the one-site binding model used for fitting may not be accurate and we thus refer to the affinities as “apparent K_D values” in the text to indicate this uncertainty.

Response Figure R5. Preparation of 2A-40S complexes for cryo-EM. **a**, Size-exclusion chromatogram of purified 2A on a Superdex 6 3.2/300 column. 2A is only detectable by immunoblot in the late-eluting peak and not in early fractions. **b**, as in a, following the incubation of 2A with purified RRL 40S subunits. 2A co-migrates with the 40S peak as confirmed by immunoblot, indicative of binding.

We attempted to prepare 2A-40S complexes for cryo-EM studies. Analysis by size-exclusion chromatography revealed that 2A co-eluted with the 40S peak (**Response Figure R5**) but despite extensive optimisation, subsequent cryo-EM imaging was unsuccessful due to issues with sample aggregation. However, the 70S-2A structure is nevertheless highly valuable because prokaryotic translation systems are well-established models for studying the detailed kinetics of -1 PRF, including at eukaryotic signals (e.g. see PMID: 24919156, 24949973, 31911945 and 32427100). Binding of 2A at a location that may compete with elongation factor binding is therefore an important result. Additionally, our structure definitively reveals the RNA-binding mechanism of 2A (a novel fold), and this proposed mechanism is consistent with our subsequent biochemical validation performed using eukaryotic ribosomes (see **Figure S5**, in particular panel f).

“Although the location of 2A binding sites suggest that 2A might interfere with binding of elongation factors (and thus lead to a kinetic delay in elongation that might enhance PRF), the authors were not able to distinguish between the relative importance of 2A’s binding to the viral mRNA and to ribosomal subunits in mediating PRF. They note that binding of 2A to the ribosome and to viral mRNA involves the same elements of 2A and would thus be mutually exclusive. Indeed, saturation of ribosomes with 2A late in infection might contribute more to a shut-off of translation than to PRF.”

As discussed in the response to reviewer 2 (above), -1 PRF efficiency is primarily determined by the ability of 2A to bind the stimulatory element rather than the ribosome, and this point has been clarified in the discussion.

“In conclusion, this manuscript reports several important advances regarding a mechanism of protein-dependent PRF that the authors have established as a paradigm for this type of process. The results are significant, clearly presented and likely to be of general interest.”

We thank the reviewer for this positive overall appraisal of our work.

“Specific comment.

1. Lines 822-823. The description of the method used for purification of ribosomal subunits from rabbit reticulocytes includes an incorrect reference. This cited paper refers to two relevant papers, both of which concern E. coli ribosomes rather than mammalian ribosomes.”

We apologise for this error, this has been corrected to the following reference:

“Pisarev, A.V., Unbehauen, A., Hellen, C.U. & Pestova, T.V. Assembly and analysis of eukaryotic translation initiation complexes. Methods Enzymol 430, 147-77 (2007).”

Reviewer #4 (Remarks to the Author):

“The manuscript by Hill et al. addresses the mechanism underlying the function of a viral protein, 2A. The work includes the crystal structure determination of 2A, interaction analysis with mRNA regions relevant for programmed -1 ribosomal frameshifting, complex formation with the small ribosomal subunit and cryo-EM structure determination of a bacterial 70S translation initiation complex with 2A bound, complemented with single molecule analysis. This impressive integrated study hence provides a number of novel findings, which are overall clearly presented.”

We thank the reviewer for these kind comments.

“Nevertheless, some parts would benefit from a clearer discussion. For example, it is not clear why there should be multiple binding sites for 2A on the ribosome (is that not just an effect of using a large excess of 2A for the biochemical reconstitution of the complex?), while the primary target described as being selective is the viral mRNA. Also, as 2A targets eukaryotes, how relevant is it to use a bacterial ribosome to reconstitute a complex with 2A? That frameshifting works in both species doesn't explain this per se, the binding sites could still be different.”

As discussed in response to reviewer 2 (above), we believe that the 2A molecule recognises specific RNA conformations that include both the mRNA (presumably folded as a pseudoknot) and a site on the ribosome surface (bound by 2A1, **Figure 5e**). However, we observe conformational plasticity of 2A that we believe facilitates its binding to other, sub-optimal RNA conformations that are none the less occupied in the cryo-EM structure given the significant molar excess of 2A used to prepare the grids (3.0 μ M 2A vs. 75 nM ribosomes). We have added a sentence to the discussion to indicate that 2A2 and 2A3 may be ‘low affinity’ sites and have also added a note on the conservation of the 2A1 binding site between bacterial and eukaryotic ribosomes (**Figure S7**, see also responses to reviewer 3 above).

“What is the meaning of the covalently bound dimer of 2A in the context of mRNA recognition or presence on the ribosome, there seem to be no dimers involved there; this could be discussed.”

This is likely not relevant to either mRNA recognition or ribosome binding. Regarding the former, the C111S mutation has no significant effect on RNA binding as judged by EMSA (**Figure S2a**). Regarding the latter, C111 is reduced and no disulfide bridges are seen between any 2A molecules

in our cryo-EM structure. Instead, we suggest that this is an artifact of crystallisation (which consistently took > 30 days, even when optimised). We now mention this in results. As an aside, it wasn't until we solved the structure that we realised that the DTT in the crystallisation buffer may have been preventing lattice formation! Given the very good agreement between our crystal structure and cryo-EM structure, there is no suggestion that this unusual crystal contact changes the native fold of 2A in any way.

“Detailed points:

- thank you for sharing maps and atomic models. The crystal structure looks fine; minor point: what is the meaning of the Arg109 –SO4²⁻ ion? Sounds non-specific.”

It probably originates from the crystallisation buffer, which contained 0.625 M ammonium sulfate. We modelled it based on unexplained density close to R109 in the F_o-F_c maps, but we agree that it likely has no physiological relevance and have thus refrained from discussing it in detail in the text.

“Cryo-EM map: the refinement of the atomic model looks good. However, the contour level needs to be adjusted rather strongly between the different 2A molecules. This is a sign that the stoichiometry is rather variable, possibly answering the question of functional relevance of these, i.e. in fact there might be only one really relevant. It is surprising to see several 2A molecules anyway, is the RNA fold recognition similar? A super-position may help (see point below)... The density is good for one, but hardly resolves the secondary structure of the others. This should be clarified in the text to not give the impression of 3 (or 4?) well defined molecules; again, this suggest that one is well bound. What is the excess used to prepare the complex? This should be discussed in this context.”

We thank the reviewer for pointing this out, see further discussion below. Regarding fold recognition and superposition, see **Response Figure R1 and R4** above. As mentioned above, there was a ~40-fold molar excess of 2A used to prepare grids (3.0 μM 2A vs. 75 nM ribosomes). We agree with the reviewer that, based on the quality of the density, 2A1 is likely the highest affinity binding site, and likely most relevant. The functional relevance of 2A2, 2A3 and 2A4 are unclear, and we have added a sentence to the discussion to reflect this.

“Finally, there is a large two-domain density between the 30S beak and the 50S ribosomal subunit, which is not accounted for – yet another molecule?”

Despite trying additional classification schemes, we were unable to resolve this additional density (even at low resolution) so we refrain from speculation in the text. It is likely flexible and heterogeneous with respect to the three/four copies of 2A that we do discuss. It is possible that it could represent an additional partial-occupancy 2A molecule, components of the L7/L12 stalk, part of the downstream RNA stimulatory element (which is present in the reporter mRNA used to initiate the ribosomes) or a combination of all of the above.

“- crystal structure of 2A: MALS seems to indicate a monomer, but the structure shows a covalently bound dimer. Are there any such dimers on the viral mRNA or on the ribosome (it doesn't seem so)?”

No, as discussed above, we do not see such dimers and believe the covalent dimers observed *in crystallo* to an artefact of crystallisation with no direct physiological relevance.

“- as there are 4 molecules in the ASU with NCS, what are the structural differences (only briefly mentioned) and what could these tell for 2A function?”

The structural differences observed between the four NCS-related molecules in the ASU are small (following successive pairwise alignment of chains, overall mean C α RMSD = 0.34 Å). There are only two regions of significant conformational variability. The first is the ‘arginine loop’ - which makes functional sense because the cryo-EM structure shows how this can adopt different conformations to recognise different RNA targets. The second is the β 2-loop- β 3 region, and the functional consequences of this are unclear, as it is distal to the RNA-binding surface. We have highlighted these conformational differences in **Figure 1f** and we make reference to the conformational flexibility of the key ‘arginine loop’ in the context of 2A binding similar-but-distinct RNA structures in the context of the 70S ribosome.

“- line 130: binding of 2A to EMCV: should probably say EMCV mRNA”

Corrected, thanks.

“- pages 6 and 7 on single molecule measurements: is a relatively lengthy section with a number of hypotheses but not that conclusive, could be more concise”

We appreciate the reviewer’s comment but feel that the length is justified by the complexity of the data, which has been re-analysed following the comments of reviewer 2.

“- line 265: the hypothesis that 50S ribosomal subunit joining and 70S IC formation is compatible with the presence of 2A should be carefully analysed. What would it mean that 2A remains bound in a 70S IC complex? When would it dissociate? e.g. in a study a few years ago among the initiation factors IF1, IF2 and IF3, IF3 had been proposed to remain bound, but that turned out later to be incorrect (an rRNA segment had been interpreted as IF3). In this context it is also not clear what the multiple binding sites of 2A mean, are they all relevant?”

This statement was intended to reflect the fact that we observe this complex both by MST in **Figure 4** and cryo-EM in **Figure 5**. The reviewer is correct to point out that we have no data to suggest if or when 2A would dissociate, but we note that the highest-affinity site (2A1) would clash with EF-G, so it is unclear to what extent a 2A-liganded IC would be translationally competent. 2A1 would also clash with IF2 binding, so it is unclear to what extent it would affect initiation or the transition to elongation. We have therefore removed the statement that “50S joining and initiation complex assembly is compatible with the 2A:30S interaction”. Future work on the detailed kinetics of 2A and EF-G binding in reconstituted systems may help to resolve this ambiguity.

With regard to multiple binding sites, please see also above response and response to reviewer 3.

“- line 273: how do the 7% or 15% (on a bacteria-optimized mRNA linker length) relate to the amount of frameshifted protein needed / produced in cellulo by the virus?”

This is lower -1 PRF efficiency than that seen in the context of virally infected cells. Using ribosome profiling, we have previously estimated this to reach a maximum of ~70% at later timepoints during infection (PMID: 28593994). However, *in vitro* the efficiency of -1 PRF in rabbit reticulocyte lysate is ~20%, and our reported *in vitro* efficiency in bacterial systems is similar to this value. The reason for this difference between *in vivo* and *in vitro* efficiencies is unknown, but it could reflect additional

effects of distal RNA sequences, presence of viral effector proteins, or differences in translational dynamics (i.e. monosomes vs polysomes).

“- what is the meaning of 3 or 4 2A copies bound on the ribosome? Are these significant? A structural comparison / superposition might help. Are the binding regions conserved between prokaryotic and eukaryotic ribosomes? If not, what is the meaning to have done the structural analysis on a bacterial ribosome as the primary target of the virus is not bacterial apparently?”

See above response to reviewer 3, and **Response Figure R4 and R5** above.

“- line 381: EF-G “catalysed”: as translocation is not an enzymatic step by itself it would be better to call this “mediated”; in fact, translocation can happen even without EF-G, but at slower rates”

Corrected, thanks.

“- legend Fig. 5: the term is “focused” classification and refinement, e.g. see references such as Curr Op Struct Biol 2017, eLife 2018, Biosci Rep 2018 etc.; to be corrected throughout, e.g. Suppl. Fig. 6 etc.”

Done, changed throughout.

“- legend Fig. 5: the term cryo-EM “electron” density is technically incorrect. Due to the nature of electrons, which are charged, these are not electron density maps as in X-ray crystallography. Instead, these are electrostatic potential maps (see e.g. Wang & Moore, 2017; Hryc et al., 2017; Marques et al., 2019; Wang et al., 2021). A common way of saying would be simply “cryo-EM map”. “

Done, changed throughout.

“- Fig. 6: is the stem-loop structure a hypothesis / structure prediction from mRNA fold prediction? If so, it should be mentioned, especially as the pseudoknot structure now comes from this study”

Yes, this is a predicted structure. We had originally described this in the “RNA folding prediction” section of Methods, but we have now additionally clarified this in the Fig 6 legend.

“- it is good that the structures and maps will be deposited. For Table 1, Rmerge is an outdated value and can/should be removed”

Done.

“- line 949: instead of “initial model” a better term would be “initial reference””

Changed, thanks.

“- line 958: CTF-refinement gave 2.5 Å resolution, but the final reconstruction is at 2.66 Å, why is that? Which map provides more details or less noise?”

The 2.66 Å map provides better density and clearer details in the vicinity of the 2A binding site. This was generated from the fully rescaled, CTF-refined particles following an additional round of focused classification and refinement. We have added a comment on this point to the methods.

REVIEWERS' COMMENTS

Reviewer #1 (Remarks to the Author):

*I had no problems with this ms before, I have none now either.

Reviewer #2 (Remarks to the Author):

The authors have provided thorough and thoughtful responses to the comments from the reviewers. I believe the authors have satisfactorily addressed all of the reviewers' concerns, with the exception of some items that are technically not practical to solve.

Two relatively minor issues arising from some of the new single-molecule analysis remain to be resolved:

1) Many of the numerical values presented in the manuscript are listed with errors, but some are not (notably the energetic stability values). In cases where a quantitative value is being reported (rather than an approximate value), it is most appropriate to list the experimental uncertainty.

2) For the calculation of thermodynamic stabilities from pulling curves using the Jarzynski Equality, it is unclear if the authors have taken into account the intrinsic bias in the Jarzynski estimator: whereas the Crooks Fluctuation Theorem provides an unbiased estimate of the free energy, the Jarzynski Equality always overestimates the free energy for any finite-size experimental sample. This issue is discussed, for example, in Gore et al PNAS 100:12564 (2003) or Palassini et al Phys Rev Lett 107:060601 (2011). I would guess that the correction is probably about 1-2 kcal/mol for these measurements, depending on how far out of equilibrium the system is. The authors can use the equations in Gore et al to obtain a reasonable estimate of the bias that will need to be subtracted. Note that this bias correction increases the error in the estimate of the stability of the structure.

Reviewer #3 (Remarks to the Author):

The authors have responded in a satisfactory manner to my comments and questions. However, the link to Mendeley data does not appear to be active, so that the completeness of the primary data deposited there could not be evaluated.

Reviewer #4 (Remarks to the Author):

Thanks for having taken into account comments & suggestions.

No further comments from my side.

Reviewer #1 (Remarks to the Author):

"I had no problems with this ms before, I have none now either."

Great, thanks.

Reviewer #2 (Remarks to the Author):

"The authors have provided thorough and thoughtful responses to the comments from the reviewers. I believe the authors have satisfactorily addressed all of the reviewers' concerns, with the exception of some items that are technically not practical to solve.

Two relatively minor issues arising from some of the new single-molecule analysis remain to be resolved:

1) Many of the numerical values presented in the manuscript are listed with errors, but some are not (notably the energetic stability values). In cases where a quantitative value is being reported (rather than an approximate value), it is most appropriate to list the experimental uncertainty.

We have now added 5% standard errors to the mfold predicted energy values as suggested on the mfold website, and we have ensured that all the reported values contain also the uncertainty information.

2) For the calculation of thermodynamic stabilities from pulling curves using the Jarzynski Equality, it is unclear if the authors have taken into account the intrinsic bias in the Jarzynski estimator: whereas the Crooks Fluctuation Theorem provides an unbiased estimate of the free energy, the Jarzynski Equality always overestimates the free energy for any finite-size experimental sample. This issue is discussed, for example, in Gore et al PNAS 100:12564 (2003) or Palassini et al Phys Rev Lett 107:060601 (2011). I would guess that the correction is probably about 1-2 kcal/mol for these measurements, depending on how far out of equilibrium the system is. The authors can use the equations in Gore et al to obtain a reasonable estimate of the bias that will need to be subtracted. Note that this bias correction increases the error in the estimate of the stability of the structure."

Following the reviewer's suggestion, we now used the workflow by Gore *et al.* to estimate the bias in the calculations and accordingly we have updated the values (**Supplementary Table 3**). Indeed, for both, WT and CUC without 2A, and the CUC with 2A, the bias correction was rather small (≤ 1 kcal/mol). This is in agreement with the highly overlapping unfolding and refolding work distributions that were used to estimate the Gibbs free energy using the Crook's theorem. However, in case of WT+2A, due to the 2A-induced stabilization, the system is further away from the equilibrium. Therefore, also the estimated bias was higher (≈ 4.7 kcal/mol). We have updated the results text to the new values, and have added a sentence to Methods with a citation of the Gore *et al.* paper.

Reviewer #3 (Remarks to the Author):

“The authors have responded in a satisfactory manner to my comments and questions. However, the link to Mendeley data does not appear to be active, so that the completeness of the primary data deposited there could not be evaluated.”

We have now replaced this broken link and published the dataset [<http://dx.doi.org/10.17632/gkpwngy65h.2>]. We have also included all raw data in the **Source Data** file.

Reviewer #4 (Remarks to the Author):

“Thanks for having taken into account comments & suggestions. No further comments from my side.”

Great, thanks.

Editorial Request:

Figure 5 is too large (too "busy"). Please divide it into two separate figures.

Thanks for this feedback. We have now divided the old **Figure 5** into three figures, each with a clear, independent take-home message. This greatly improves the clarity of presentation. We have also removed one panel entirely: **Fig 5d**. This used to show the cryo-EM density for sidechains. Whilst this used to be common for cryo-EM structures, it is not biologically informative and may actually be misleading – as now discussed in response to previous comments of reviewer 4, the cryo-EM density is not of equivalent quality for all three copies of 2A.

Please see overleaf for the new figures, reproduced here for convenience.

Figure 5. 2A binds to the 70S ribosome via interactions with the 16S rRNA. **a**, Cryo-EM analysis of a complex formed between initiated *E. coli* 70S ribosomes and EMCV 2A. Images ($\times 75,000$) were recorded on a Titan Krios microscope. Representative micrograph from dataset of 5730 images. **b**, Cryo-EM map at 2.7 Å resolution after focused classification and refinement. Three copies of 2A (orange, red, yellow) are bound to the 16S rRNA of the small (30S) subunit (blue ribbon). **c**, Close-up view of the 2A binding site. Ribbon diagrams of 2A (coloured as above) and ribosomal RNA (purple) are shown. Protein N- and C- termini are labelled. **d**, Superposition of the three copies of 2A reveals a common RNA-binding surface with conformational flexibility. Residues involved in rRNA binding are labelled and shown as sticks.

Figure 6. The 'arginine loop' plays a central role in RNA recognition. a-c, Details of rRNA recognition by 2A. For each copy of 2A, selected residues involved in interactions are labelled and shown as sticks <Insets> View of the rRNA surface bound by each copy of 2A. The rRNA helices are colour-coded and labelled. The 2A contact surface is shown as a coloured mesh (orange, red and yellow, respectively). d-f, Close-up view of interactions between the 2A 'arginine loop' residues (R95, R97 and R100) and the rRNA backbone (sticks) for each copy of 2A (orange, red, yellow). Polar or electrostatic contacts are indicated by a green dashed line.

Figure 7. 2A binding may clash with translational GTPases. **a**, Ribbon diagram of initiated 70S-mRNA-tRNA^{fMet}-2A complex. Ribosome sites are labelled A, P and E. The initiator tRNA^{fMet} (dark green), mRNA (light green), and 2A (orange, red, yellow) are shown in two orthogonal views. **b**, Comparison of 70S-2A complex to 70S pre-translocation complex with EF-G (4V7D). 2A binding would clash (red wedges) with EF-G binding. **c**, Comparison of 70S-2A complex to 70S complex with EF-Tu (5WE6). 2A binding would clash (red wedges) with EF-Tu binding.